# The Smc5/6 complex counteracts R-loop formation at highly transcribed genes in cooperation with RNase H2

Shamayita Roy, Hemanta Adhikary, Sarah Isler, Damien D'Amours*

Ottawa Institute of Systems Biology, Department of Cellular and Molecular Medicine, University of Ottawa, Ottawa, Canada

**Abstract** The R-loop is a common transcriptional by-product that consists of an RNA-DNA duplex joined to a displaced strand of genomic DNA. While the effects of R-loops on health and disease are well established, there is still an incomplete understanding of the cellular processes responsible for their removal from eukaryotic genomes. Here, we show that a core regulator of chromosome architecture —the Smc5/6 complex— plays a crucial role in the removal of R-loop structures formed during gene transcription. Consistent with this, budding yeast mutants defective in the Smc5/6 complex and enzymes involved in R-loop resolution show strong synthetic interactions and accumulate high levels of RNA-DNA hybrid structures in their chromosomes. Importantly, we demonstrate that the Smc5/6 complex acts on specific types of RNA-DNA hybrid structures in vivo and promotes R-loop degradation by the RNase H2 enzyme in vitro. Collectively, our results reveal a crucial role for the Smc5/6 complex in the removal of toxic R-loops formed at highly transcribed genes and telomeres.

## Editor's evaluation

This study presents an important finding showcasing the role of Smc5-6 complex in counteracting R-loops at transcriptionally active sites. The evidence supporting the claims of the authors is solid, although inclusion of a genome-wide R-loop detection assay would have strengthened the study. The work will be of interest to scientists studying genome structure and stability.

*For correspondence:
damien.damours@uottawa.ca

**Competing interest:** The authors declare that no competing interests exist.

## Introduction

The maintenance of genome stability is a primordial function that ensures the proper development and homeostasis of all living organisms (*Negrini et al., 2010*; *Hanahan and Weinberg, 2011*; *Lengauer et al., 1997*; *Cifone and Fidler, 1981*). Successful maintenance of genomic integrity requires constant monitoring and repair of DNA lesions because genomes are under constant attack from endogenous sources of DNA damage (*Tubbs and Nussenzweig, 2017*; *Thada and Greenberg, 2022*; *Lindahl and Nyberg, 1972*). One of the most common sources of endogenous DNA damage is the formation of RNA-DNA hybrid structures in chromosomes. During transcription, nascent RNA transcripts can re-anneal to complementary DNA strands producing an RNA-DNA hybrid and displace the template strand, creating an obstacle to the progression of the DNA replication machinery. The resulting RNA-DNA hybrid and displaced ssDNA segment —together termed the R-loop— are highly deleterious for genome integrity (*Chatzidoukaki et al., 2021*; *Costantino and Koshland, 2018*; *De Magis et al., 2019*; *Stork et al., 2016*). RNA-DNA hybrid structures can also be formed under a variety of physiological conditions in eukaryotic genomes and play important roles in cell physiology and regulate genome dynamics. Common physiological roles of

**eLife digest** Cells are constantly exposed to external and internal processes that threaten the integrity of their genetic information. Even small errors or physical defects in the DNA strands that store the instructions required for life can have wide-ranging consequences. In response, cells deploy a variety of molecular actors to repair these lesions before they cause issues.

R-loops are a particularly complex form of genetic damage that emerge when a molecule known as RNA inserts itself into the DNA helix. If left unaddressed, the resulting structure interferes with the machinery that allows cells to replicate or express their genes – potentially leading to serious harm for the organism. Indeed, R-loops are often present in genes linked to a variety of cancers and neurological disorders. Despite their importance, how R-loops are normally removed is still not fully understood.

To explore this question, Roy et al. focused on the Smc5/6 complex, a molecular machine found across the tree of life that can help repair structurally complex genetic lesions. The team tested its involvement in R-loop removal using yeast strains that tend to carry more of these defects.

Yeast cells genetically manipulated to lack functional Smc5/6 complexes accumulated toxic levels of R-loops, resulting in extreme growth defects. Further investigations showed that the complex particularly targeted R-loops forming in highly expressed genes, as well as in genetic sequences important for preserving DNA integrity. Finally, Roy et al. used biochemical assays to explore how the human Smc5/6 complex specifically recognizes R-loops and assists an enzyme called RNase H – which can degrade RNA – in removing them.

Taken together, these findings deepen our understanding of R-loop dynamics; going forward, they may also help explain why mutations in components of the Smc5/6 complex are associated with severe genetic disorders in humans.

RNA-DNA hybrids include immunoglobulin class switching recombination of B cells in vertebrates (*Yu et al., 2003*; *Roy et al., 2008*), mitochondrial DNA replication (*Pohjoismäki et al., 2010*; *Xu and Clayton, 1996*), bacterial plasmid replication (*Baker and Kornberg, 1988*; *McLean et al., 2022*), and CRISPR-Cas9 gene editing (*Zhang et al., 2021*; *Xiao et al., 2017*). Moreover, RNA-DNA hybrids co-ordinate specific regulatory steps in transcription initiation and termination (*Sidorenkov et al., 1998*; *Skourti-Stathaki et al., 2011*; *Nudler et al., 1997*), telomere homeostasis (*Balk et al., 2013*; *Luke et al., 2008*), and gene expression (*Crossley et al., 2019*; *García-Muse and Aguilera, 2019*; *Brambati et al., 2020*; *Niehrs and Luke, 2020*; *Zardoni et al., 2021*). However, the formation of unprogrammed or non-physiological RNA-DNA hybrid structures can interfere with DNA replication-related processes, resulting in replicative stress and the formation of DNA double-strand breaks (DSBs) (*Crossley et al., 2019*; *Aguilera and García-Muse, 2012*; *Santos-Pereira and Aguilera, 2015*; *Hamperl et al., 2017*; *Kumar et al., 2021*; *Kim et al., 2024*). Importantly, exposed ssDNA in the R-loop can be cleaved by different endonucleases leading to DNA breaks and/or mutagenic events, and can also adopt harmful secondary structures (*Freudenreich, 2018*; *Miglietta et al., 2020*). R-loop-induced DNA damage and genomic rearrangements have been linked to various disease states in humans. Examples include trinucleotide repeat-associated diseases, auto-immune disorders, neurological disorders, and cancer, although it is not currently known whether R-loops play a causative or consequential role in such diseases (*García-Muse and Aguilera, 2019*; *Richard and Manley, 2017*).

To mitigate the toxic consequences associated with the presence of unprogrammed RNA-DNA hybrids in chromosomes, several cellular mechanisms work in concert to prevent their formation, and when they do accumulate, remove them from eukaryotic genomes. For instance, the Ribonuclease H (RNase H) family of enzymes plays a central role in degrading the RNA moiety of R-loops created under a variety of genomic conditions (*Lazzaro et al., 2012*; *Lockhart et al., 2019*; *Figure 1A*). Recently, it was shown that the RNase DICER can also cleave the RNA moiety of the R-loops in higher eukaryotes (*Camino et al., 2023*). In addition, factors associated with transcription and mRNA biogenesis, RNA-DNA helicases, topoisomerases, chromatin remodelers, and several DNA repair enzymes are known to be involved alongside RNase H in preventing the accumulation of R-loop structures in eukaryotic chromosomes (*García-Muse and Aguilera, 2019*). However, a definitive understanding of

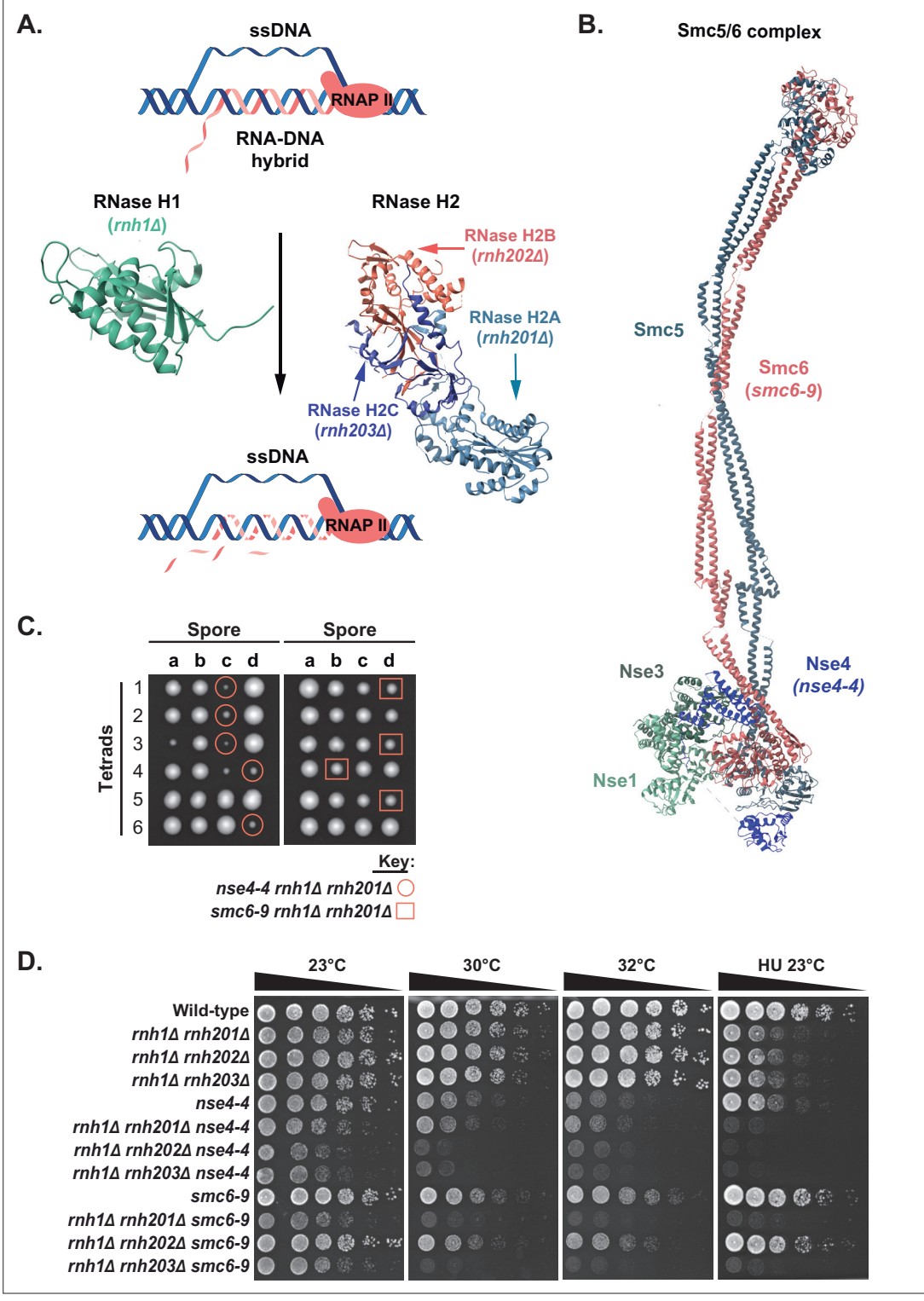

**Figure 1.** Synthetic enhancement of RNase H enzyme defects by mutations affecting Smc5/6 complex activity. (**A**) Schematic model showing the mechanism of RNase H mediated RNA-DNA hybrid degradation. The crystal structures of the RNase H1 and RNase H2 complex used in this representation are PDB: 2QK9 (***Nowotny et al., 2007***) and PDB: 3PUF (***Figiel et al., 2011***), respectively. (**B**) Schematic representation of the Smc5/6 complex showing its subunits and the corresponding mutant alleles used in this study. The crystal structure of the Smc5/6 complex used in this representation is PDB: 7QCD (***Hallett et al., 2022***). (**C**) Growth of *nse4-4 rnh1Δ rnh201Δ* and *smc6-9 rnh1Δ rnh201Δ* haploid spores after sporulation and germination of heterozygous diploid strains at

*Figure 1 continued on next page*

*Figure 1 continued*

23 °C. The viability of the haploid spores was scored after 3 days of germination. (**D**) The proliferation capacity of combination mutants affecting Smc5/6 complex and RNase H activity was monitored after dilution on solid medium and growth under various conditions (indicated on top of the growth medium). Concentration of HU used was 12.5 mM. YPD 23 °C, 30 °C, 32 °C, and HU plates were grown in temperature-controlled incubators for ~48 hr, ~28 hr, ~26 hr, and ~72 hr respectively, before scanning the plates.

The online version of this article includes the following figure supplement(s) for figure 1:

**Figure supplement 1.** Growth phenotype of yeast strains carrying mutations affecting RNA-DNA hybrid metabolism enzymes and SMC5/6 complex components.

the sequence of events and exact molecular mechanisms responsible for the repair of R-loops/RNA-DNA hybrids remains to be established.

Work from our laboratory and other researchers provides hints of a possible involvement of structural maintenance of chromosomes (SMC)-type complexes in RNA-DNA hybrid metabolism (*Lafuente-Barquero et al., 2017*; *Serrano et al., 2020*; *Girasol et al., 2023*; *Penzo et al., 2023*). SMC complexes are effectors of large-scale changes in chromosome organization and include three evolutionarily conserved enzyme complexes: condensin, cohesin, and the Smc5/6 complex (*Peng and Zhao, 2023*). The Smc5/6 complex is a particularly intriguing member of this family because its function —unlike that of cohesin and condensin— is primarily concerned with DNA repair, and yet its exact contribution to this process is not fully understood. Recent evidence suggests that the Smc5/6 complex is a DNA compacting enzyme that acts in vivo by regulating local chromatin domains containing unusual DNA structures that can lead to replication stress and/or DNA damage (*Serrano et al., 2020*; *Gutierrez-Escribano et al., 2020*; *Tanasie et al., 2022*; *Pradhan et al., 2023*). Interestingly, genome-wide screens as well as targeted genetic analyses revealed synthetic interactions among a subset of mutants affecting Smc5/6 complex components and RNA-DNA hybrid detoxification enzymes (*Lafuente-Barquero et al., 2017*; *Costanzo et al., 2016*; *Styles et al., 2016*; *Kuzmin et al., 2018*; *Chang et al., 2019*). Consistent with this, we have previously shown that the Smc5/6 complex can bind to short RNA-DNA duplexes with high affinity and specificity (*Serrano et al., 2020*). Together, these results raise the intriguing possibility that the Smc5/6 complex might be involved in the detection and/or processing of toxic R-loops formed in eukaryotic genomes.

Here, we show that the Smc5/6 complex is directly involved in the repair of unscheduled R-loops generated by a diverse set of genomic transactions. In particular, we demonstrate that the Smc5/6 complex binds strongly to R-loop structures formed during active gene transcription and promotes RNase H2-mediated degradation of the RNA component of R-loops. Our results unravel a hitherto unanticipated role for the Smc5/6 complex in the removal of RNA structures from chromosomes, an essential function for the maintenance of genome integrity and cell fitness.

## Results
### The Smc5/6 complex collaborates with RNase H enzymes in the maintenance of genome integrity

In eukaryotes, two partially overlapping enzymes mediate the degradation of R-loops; RNase H1 and RNase H2 (*Figure 1A*; *Lazzaro et al., 2012*; *Lockhart et al., 2019*). To determine whether the Smc5/6 complex contributes to R-loop repair in proliferating cells, we introduced the *smc6-9* and *nse4-4* alleles of the Smc5/6 complex (*Figure 1B*) into budding yeast strains defective for RNase H1 (*rnh1Δ*) and RNase H2 (*rnh201Δ/rnh202Δ/rnh203Δ*) activity. We used the *smc6-9* and *nse4-4* alleles for our analysis because they correspond to moderate and a strong temperature-sensitive mutants of the Smc5/6 complex, respectively, and inactivate its DNA repair activity in a general/non-specific manner (*Ben-Aroya et al., 2008*; *Hwang et al., 2008*). For simplicity, we will refer to strains defective in both Smc5/6 complex and RNase H activities as double mutants and their parent strains as single mutants (i.e., even if the corresponding strains carry more than one mutant allele).

The proliferation rate of the Smc5/6-RNase H double mutants was compared to that of parental single mutants at various temperatures (23 °C, 30 °C, and 32 °C) and in the presence of DNA-damaging agents (hydroxyurea [HU], 4-nitroquinoline 1-oxide [4NQO], and methyl methanesulfonate [MMS]) (*Figure 1C and D* and *Figure 1—figure supplement 1A*). Loss of both Smc5/6 and RNase

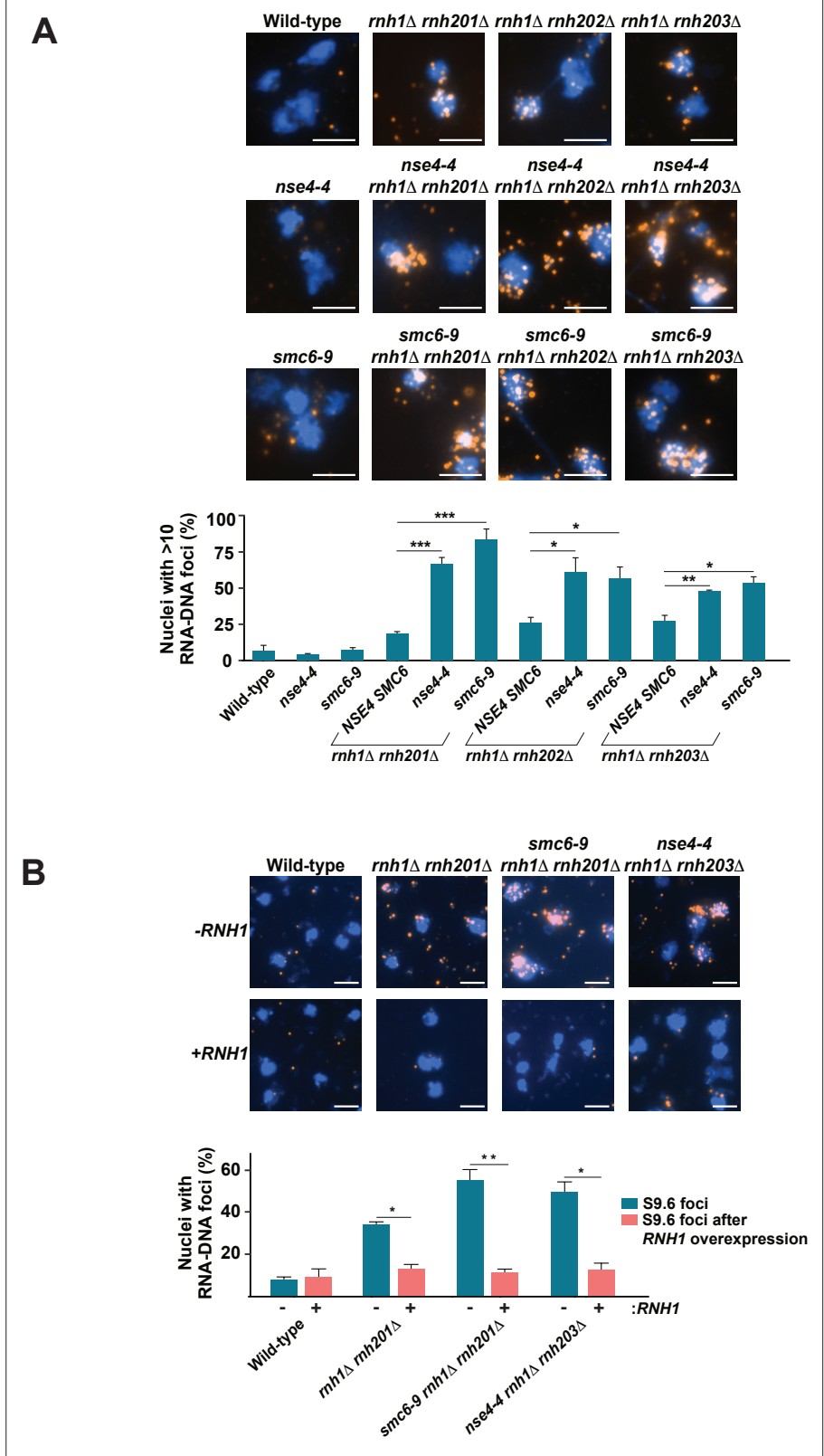

**Figure 2.** RNA-DNA hybrid accumulation in cells defective for Smc5/6 complex and RNase H activity.
(**A**) The abundance of RNA-DNA hybrids in chromosomes was monitored using the S9.6 antibody by indirect immunofluorescence microscopy on chromosome spreads prepared from wild-type (WT), single- and double-mutant yeast strains grown at 23 °C. Representative spreads are shown, with DNA stained in blue (DAPI) and

*Figure 2 continued on next page*

*Figure 2 continued*

orange foci representing RNA-DNA hybrid structures detected by the S9.6 antibody. Quantification of nuclei containing S9.6 foci (>10 foci per nucleus) is shown below the images. 100-200 nuclei were visualized and manually counted for each replicate to obtain the fraction of nuclei with detectable RNA-DNA hybrids. Data represent the mean and SE of three independent experiments. *p<0.05; **p<0.01; ***p<0.001 (Student's t-test). Scale bar, 5 µm. (**B**) RNA-DNA hybrid were monitored and quantified as described above in the presence and absence of ectopic overexpression of RNase H1 at 23 °C in strains carrying *rnh1Δ rnh201Δ, nse4-4 rnh1Δ rnh201Δ,* and *nse4-4 rnh1Δ rnh203Δ* mutations. Scale bar, 5 µm.

The online version of this article includes the following figure supplement(s) for figure 2:

**Figure supplement 1.** Direct and indirect quantification of R-loops in cells defective for Smc5/6 and RNase H activity.

H activities resulted in impaired proliferation even under optimal growth conditions, as illustrated by the growth patterns of haploid spores after sporulation and dissection of heterozygous diploid strains (*Figure 1C and D*; YPD 23 °C). Competitive proliferation assays revealed that double mutant strains exhibit heightened temperature sensitivity at 30 °C and 32 °C compared to their corresponding parental strains (*Figure 1D*). Yeast cells without RNase H activity can replicate their chromosomes normally under unchallenged growth conditions, but their replication is hampered when exposed to DNA-damaging agents or replication inhibitors (*Lockhart et al., 2019*; *Heuzé et al., 2023*). Consistent with this, most of the Smc5/6-RNase H double mutants were more sensitive than single mutant strains when exposed to genotoxic stress (e.g., see HU at 23 °C in *Figure 1D*; MMS and 4NQO in *Figure 1—figure supplement 1A*). A similar growth exacerbation phenotype was observed when Smc5/6 complex mutations were combined with RNase H2 mutations in a RNase H1 proficient background (*Figure 1—figure supplement 1B*). Loss of Smc5/6 complex E3-ligase activity in the *mms21-H202Y* mutant also gave rise to a synthetic growth defect when combined with RNase H mutations (*Figure 1—figure supplement 1C*), thereby implicating sumoylation in RNA-DNA hybrid processing. Taken together, these experiments reveal that simultaneous loss of Smc5/6 complex and RNase H activities results in severe growth defects in yeast.

## Simultaneous loss of Smc5/6 complex and RNase H activity exacerbates RNA-DNA hybrid accumulation in chromosomes

Next, we wanted to evaluate whether the growth defect observed in Smc5/6-RNase H double mutants is due to defective RNA-DNA hybrid removal. We quantified RNA-DNA hybrid foci by indirect immunofluorescence on chromosome spreads stained with the S9.6 antibody (*Bou-Nader et al., 2022*). Remarkably, we detected a large increase in the numbers of S9.6 foci on chromatin spreads prepared from double mutants compared to those from single mutant strains (*Figure 2* and *Figure 2—figure supplement 1*). Whereas RNase H mutants alone typically show a few RNA-DNA hybrid/S9.6 foci per nucleus, inactivating Smc5/6 components in this genetic background resulted in the accumulation of more than 10 foci per nucleus. We observed similar results when measuring total S9.6 RNA-DNA hybrid fluorescence intensity in multiple fields of view (*Figure 2—figure supplement 1*). The presence of S9.6 foci was highest in the *nse4-4 rnh1Δ rnh201Δ* and *smc6-9 rnh1Δ rnh201Δ* strains, consistent with the fact that *rnh201Δ* represents the deletion of the catalytic subunit of RNase H2 (*Figure 2A*). RNA-DNA hybrids accumulated to similar levels in RNase H-only mutants (see *rnh1Δ rnh201Δ, rnh1Δ rnh202Δ,* and *rnh1Δ rnh203Δ* strains carrying 3–10 S9.6 foci per nucleus; *Figure 2—figure supplement 1*). *nse4-4* and *smc6-9* mutations alone did not result in a statistically significant increase in RNA-DNA foci formation, indicating that cells possess excess R-loop processing capacity when RNase H and other alternate R-loop metabolism pathways are active. Importantly, overexpression of an ectopic copy of *RNase H1* gene ($P_{GAL1}$-*RNH1*) by galactose induction in the single and double mutant strains largely suppressed the S9.6 signal on chromatin spreads (compared to control cells without RNase H1 overexpression; *Figure 2B*). These results indicate that the foci described in *Figure 2A* are reflective of RNA-DNA hybrid formation in double mutant strains. To further confirm the accumulation of RNA-DNA hybrids in Smc5/6-RNase H double mutants, we performed a S9.6 antibody-mediated immunoprecipitation assay followed by qPCR analysis, as described in *El Hage and Tollervey, 2018*. We focused this analysis on loci that are known to be highly enriched in R-loop formation (*Figure 3A*). Consistent with the chromosome spread analysis, we observed increased RNA-DNA

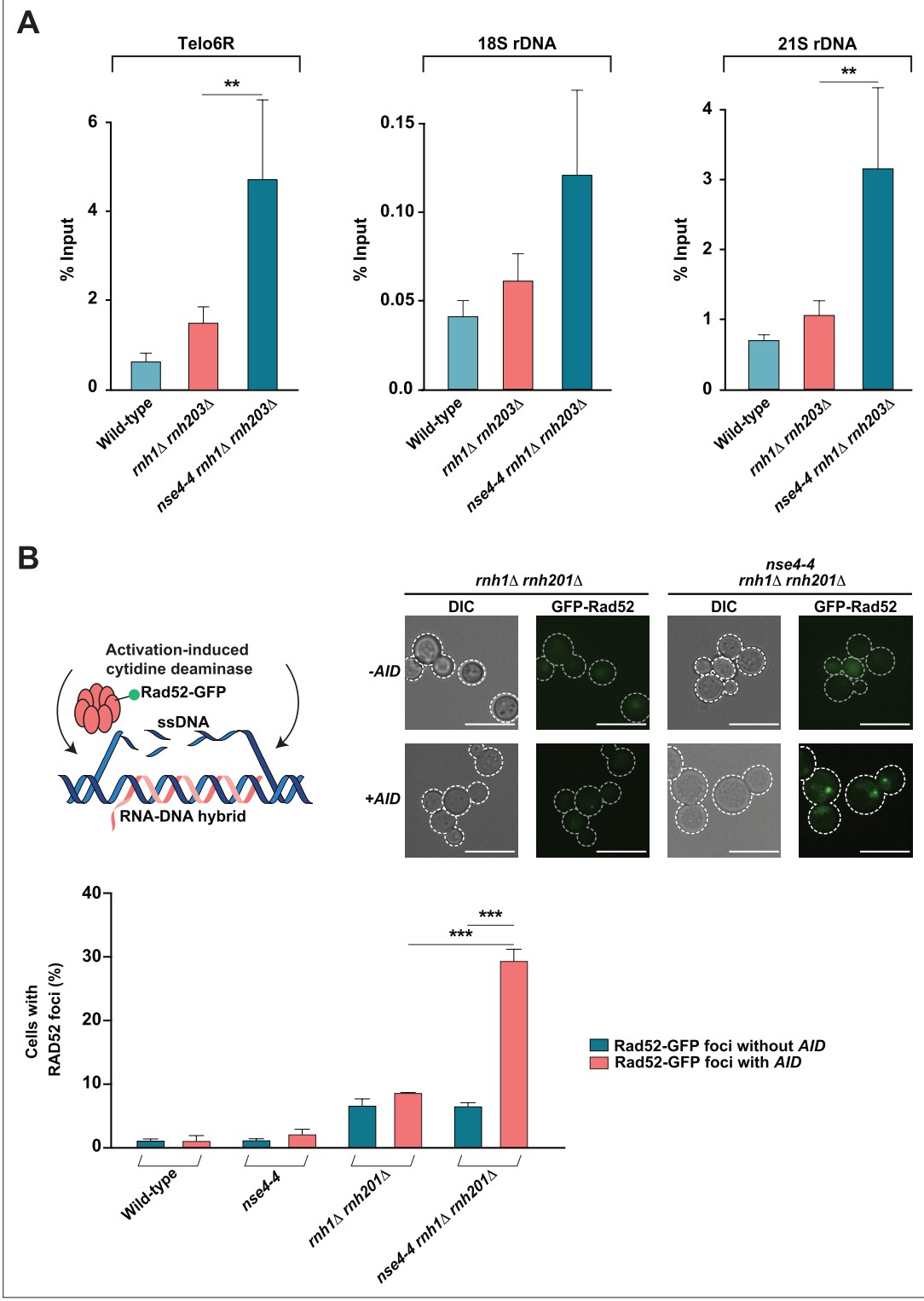

**Figure 3.** Quantification of R-loop abundance in cells defective for Smc5/6 and RNase H activity. (**A**) Mutations in the Smc5/6 complex components increase the levels of RNA-DNA hybrids at rDNA genes and telomeres in the absence RNase H activity. DRIP was performed at *TEL06R*, rDNA 18S, and rDNA 21S loci with the S9.6 antibody using genomic DNA prepared from asynchronous cultures of WT, single- and double-mutant yeast strains grown at 23 °C. Data represents the mean and SE of at least three independent experiments. *p<0.05; **p<0.01; ***p<0.001 (Student's t-test). (**B**) Detection of R-loops using AID-induced Rad52 foci formation. (Top Left) Schematic illustration of AID-induced R-loop mutagenesis and subsequent Rad52 activation. (Top Right) Representative images of cells

*Figure 3 continued on next page*

*Figure 3 continued*

carrying Rad52-GFP foci in the absence (–AID) or presence of AID (+AID). (Bottom) Quantification of cells showing Rad52-GFP foci after AID overexpression at 23 °C in WT, single- and double-mutant yeast strains. About 100 cells were visualized and manually counted for each replicate to obtain the fraction of cells with detectable Rad52-GFP foci. Data represent the mean and SE of three independent experiments. *p<0.05; **p<0.01; ***p<0.001 (Student's t-test). Scale bar, 10 μm.

The online version of this article includes the following figure supplement(s) for figure 3:

**Figure supplement 1.** Detection of R-loops using <u>a</u>ctivation-<u>i</u>nduced cytosine <u>d</u>eaminase (AID)-induced Rad52 foci formation (extended data from *Figure 3*).

hybrid immunoprecipitation specifically at telomeres and ribosomal genes in the Smc5/6-RNase H double mutant strains compared to the parental strains deficient in only RNase H activity (*Figure 3A*).

In addition to the results presented above that are based on the binding between RNA-DNA hybrids and the S9.6 antibody; we wanted to test if the accumulation of RNA-DNA hybrid structures in the Smc5/6-RNase H double mutant strain could be detected using an alternative approach. Previous studies have established that the single-stranded DNA region of R-loop structures can be directly targeted by various mutagenic enzymes, one of them being the <u>a</u>ctivation-<u>i</u>nduced cytosine <u>d</u>eaminase (AID). This enzyme is highly active on single-stranded DNA during active transcription and creates mutations in DNA by deamination of cytosine and converting cytosine into uracil. This event leads to increased Rad52 foci formation in yeast, a homologous recombination-related phenotype that can be exploited to quantify R-loop abundance upon overexpression of the AID enzyme (*Cañas et al., 2022*). In line with our previous results, we observed a substantially higher level of AID-induced Rad52-GFP foci formation in the *nse4-4 rnh1Δ rnh201Δ* mutant strain compared to a wild-type control or the parental single mutants (*rnh1Δ rnh201Δ*; *Figure 3B*, wild-type and *nse4-4*; *Figure 3—figure supplement 1*). These results suggest that the severe growth defects of yeast defective in both RNase H and the Smc5/6 complex can be linked to an accumulation of R-loops in the chromosomes of these cells.

## The Smc5/6 complex acts on R-loops formed at highly transcribed genes and telomeres

R-loops are most frequently observed at highly transcribed genes but can also be formed at telomeres and near DNA replication forks (*Figure 4A*). To identify the source of R-loops that are substrates for the Smc5/6 complex, we introduced mutant alleles of the Smc5/6 complex in yeast backgrounds that accumulate R-loops (or RNA-DNA hybrid structures) at specific genomic locations.

We first asked if unscheduled R-loops formed during active transcription are natural substrates/targets of the Smc5/6 complex. To test this notion, we used yeast strains defective for the Sen1 helicase (*sen1-1* carrying a point mutation in the helicase domain of Sen1; *Mischo et al., 2011*) and THO complex (*hpr1Δ; Luna et al., 2019*), two conditions that lead to very high levels of RNA-DNA hybrids in actively transcribed genes (*Figure 4A*). Synthetic/aggravating interactions with these two mutant conditions are frequently used as a genetic assay to test the contribution of putative effectors of R-loop metabolism (e.g., *Appanah et al., 2020*). Interestingly, deletion of *HPR1* was synthetic lethal when combined with the *nse4-4* mutation, as evidenced by the growth pattern of haploid spores following sporulation and dissection of a heterozygous diploid strain (*Figure 4B*). Although *hpr1Δ smc6-9* double mutants were viable, the growth defect associated with the *hpr1Δ* mutation was strongly exacerbated in the presence of *smc6-9* at both permissive and restrictive temperatures (*Figure 4B*). *smc6-9*, *nse4-4* and *mms21-H202Y* alleles also experienced synthetic growth defects when combined with *sen1-1* (see 30 °C /32 °C and HU/MMS conditions in *Figure 4B* and *Figure 1—figure supplement 1C*). The *nse4-4 sen1-1* mutant showed defective proliferation even under normal/unchallenged growth conditions, consistent with the more severe temperature sensitivity of this allele compared to that of *smc6-9* (*Figure 4B*). Moreover, we detected a significant increase in the numbers of S9.6 foci on chromatin spreads prepared from *nse4-4 sen1-1* and *hpr1Δ smc6-9* mutants compared to those from single mutant strains (*Figure 4—figure supplement 1*). Taken together, these genetic interactions suggest that R-loop formed at highly transcribed genes are physiological substrates for the Smc5/6 complex.

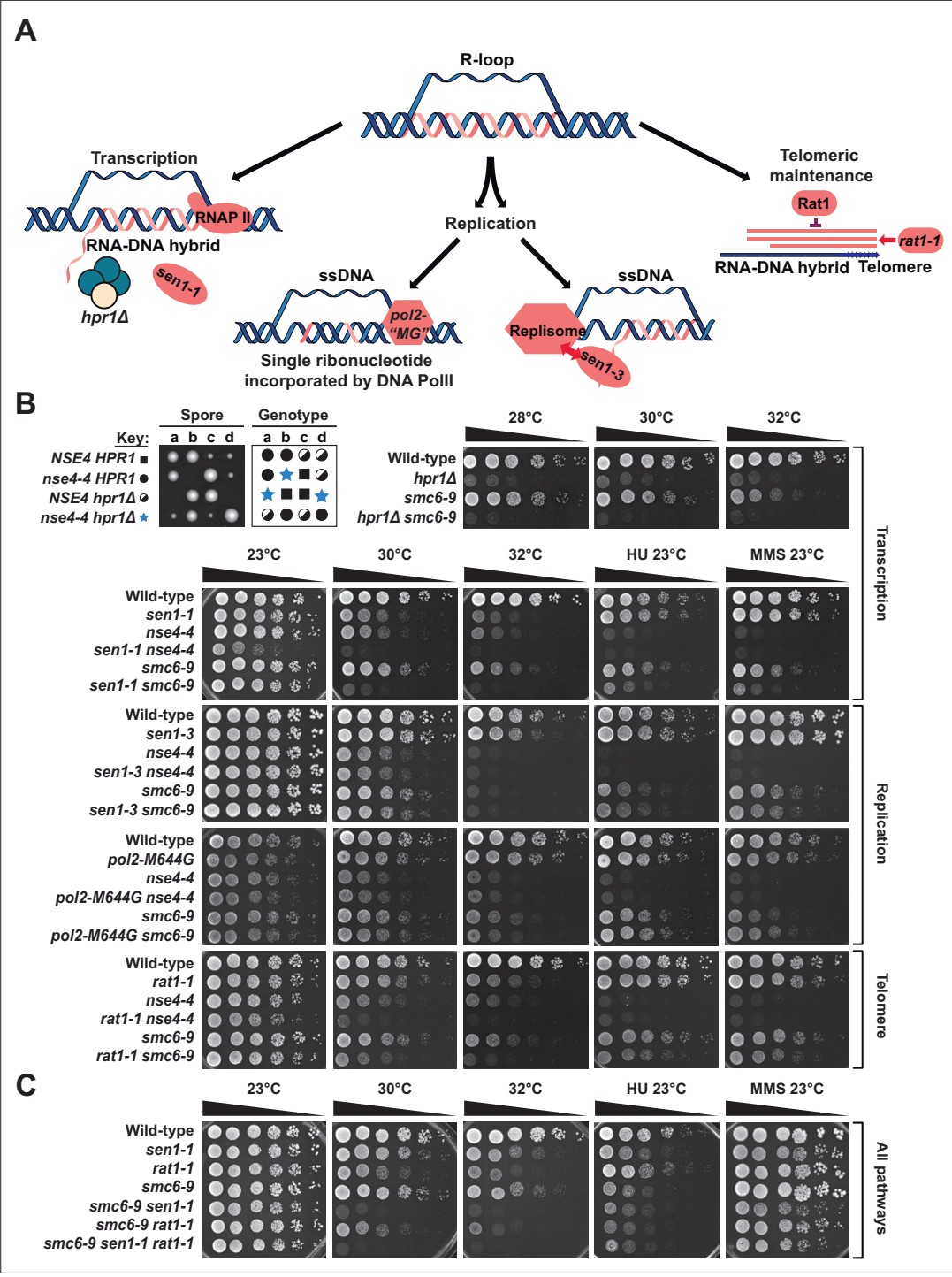

**Figure 4.** R-loops formed at highly transcribed genes and telomeres are endogenous targets for the Smc5/6 complex. (**A**) Schematic representation of various cellular mechanisms responsible for RNA-DNA hybrid formation in chromosomes and relevant proteins/mutants implicated in each process. (**B-C**) Proliferation capacity of yeast strains carrying the specified mutations was monitored by dilution assay as described in **Figure 1**. The growth temperatures and the presence of specific DNA-damaging agents (MMS concentration was 0.005%; HU concentration was 25 mM in panel (**B**) and 100 mM in panel (**C**)) in the growth medium are indicated on top of the images. YPD 23 °C, 30 °C, 32 °C, MMS, and HU plates were grown in temperature-controlled incubators for ~48 hr, ~28 hr, ~26 hr, ~48 hr, and ~72 hr, respectively, before scanning the plates.

The online version of this article includes the following figure supplement(s) for figure 4:

*Figure 4 continued on next page*

*Figure 4 continued*

**Figure supplement 1.** RNA-DNA hybrid accumulation in cells defective for Smc5/6 complex and Sen1 helicase or THO-complex activity.

Next, we combined *smc6-9* and *nse4-4* mutations with alleles of DNA polymerase ε and Sen1 helicase that increase the formation of RNA-DNA hybrids during DNA replication. Specifically, the *pol2-M644G* mutant exhibits a 10-fold increased ribonucleotide incorporation during DNA replication (*McElhinny et al., 2010*) whereas the *sen1-3* allele (carrying point mutations in the N-terminal domain of Sen1) impairs the interaction of Sen1 with the replication machinery, thereby increasing R-loop formation in the vicinity of replication forks (*Figure 4A*; *Appanah et al., 2020*). We did not observe synthetic growth defects in double mutants of these genes with *smc6-9* and *nse4-4* mutations (*Figure 4B*), suggesting that DNA replication-associated RNA-DNA hybrid structures are not substrates for the Smc5/6 complex in vivo.

Finally, we investigated whether the Smc5/6 complex interacts with a natural R-loop formed at telomeres; the <u>t</u>elomeric <u>r</u>epeat-containing <u>RNA</u> [TERRA]-DNA hybrid. To this end, we combined *smc6-9* and *nse4-4* mutations with the *rat1-1* allele defective in the 5' to 3' exonuclease activity responsible for TERRA removal (*Figure 4A*; *Luke et al., 2008*). The resulting double mutant strains exhibited stronger growth defects than the corresponding single mutants (*Figure 4B*), indicating a role for the Smc5/6 complex in R-loop metabolism at telomeres. This result is consistent with the significant accumulation of increased RNA-DNA hybrids specifically at telomeres in Smc5/6-RNase H double mutants (*Figure 3A*). Combining *rat1-1* and *smc6-9* mutations to *sen1-1* phenocopied the *sen1-1 smc6-9* double mutant (*Figure 4C*), consistent with an involvement of Sen1 in telomeric R-loops metabolism. Overall, these genetic analyses suggest that R-loops formed at highly transcribed genes and telomeres are likely physiological targets for the Smc5/6 complex.

## The Smc5/6 complex is a high-affinity R-loop-binding enzyme

The role we uncovered above for the Smc5/6 complex prompted us to investigate whether this enzyme was capable of recognizing R-loop structures directly. Extensive research over recent years has revealed that yeast and human Smc5/6 complexes are highly conserved and show strong binding affinities towards DNA substrates that mimic single-stranded (ss)–double-stranded (ds) DNA junctions, supercoiled or catenated DNA, and even branched DNA structures (*Serrano et al., 2020*; *Gutierrez-Escribano et al., 2020*; *Tanasie et al., 2022*). We have also established that the human SMC5/6 complex can bind short RNA-DNA duplexes in vitro (*Serrano et al., 2020*), suggesting the enzyme is capable of recognizing the more complex R-loop structure. To test this notion, we purified the human SMC5/6 complex and prepared R-loop and D-loop substrates (*Figure 5A–B* and *Figure 5—figure supplement 1A*) to conduct binding experiments by electrophoretic mobility shift assays (EMSAs) (*Figure 5B–D* and *Figure 5—figure supplement 1B*). The size of R-loop and D-loop substrates used in binding experiments are the same, enabling direct comparison of SMC5/6 complex affinity for these substrates (*Figure 5—figure supplement 1A*). We observed that the SMC5/6 complex can bind both R-loop and D-loop structures in EMSA experiments, but its specificity for the R-loop structure appeared slightly greater (*Figure 5C and D*). For instance, the SMC5/6 complex can bind the R-loop substrate even at a lower concentration of 6.25 nM and 12.5 nM, as evident from the protein/DNA band shift formed at the top of the gel at that concentration of enzyme (lane 2 and 3; *Figure 5C*). In contrast, we observed little to no binding to D-loops at 6.25 nM and 12.5 nM of SMC5/6 complex (lane 2 and 3; *Figure 5D*) and visibly moderate binding at the higher concentrations of protein. Consistent with this, the calculated equilibrium dissociation constant ($K_D$) of the Smc5/6 complex was slightly lower for R-loop substrates than for D-loops, but the significance of this difference, if any, remains to be established (*Figure 5—figure supplement 1C*). The similar range of binding affinities of the SMC5/6 complex for the R-loop and D-loop suggests that the single-stranded DNA part of the R-loop may be important for SMC5/6 complex binding on these structures—a mechanism previously observed for DICER-mediated R-loop resolution (*Camino et al., 2023*). Importantly, we observed that the SMC5/6 complex maintained its binding to R-loops even when challenged with 100-fold excess double-stranded DNA analogue (i.e., Poly[d(I-C)]), a behavior not observed with D-loops (lane 11–13 *Figure 5C* vs lane 10–12 *Figure 5D*). These data suggest that the binding of the Smc5/6 complex to R-loops is more resilient to challenge by a competitor substrate than the interaction of the enzyme to

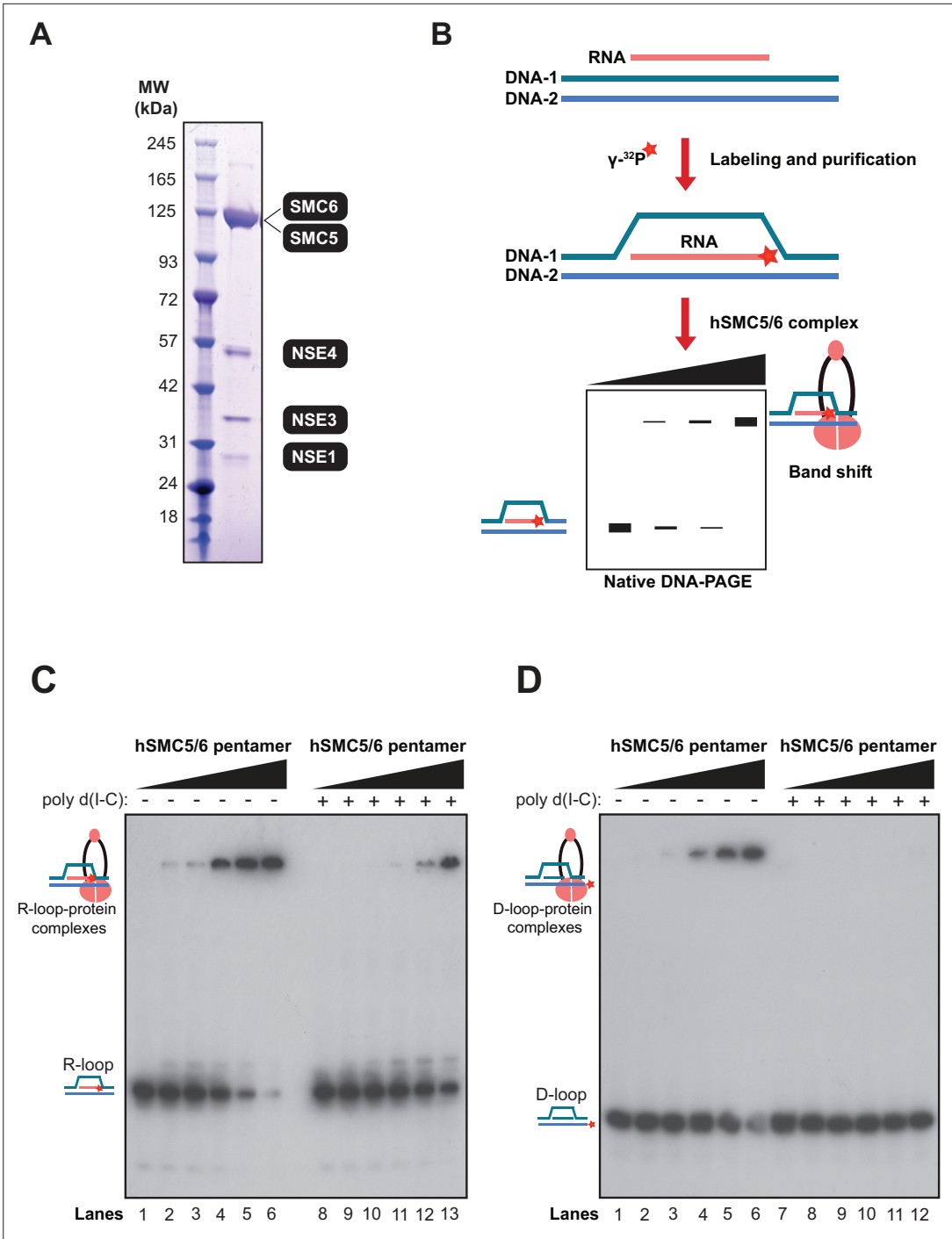

**Figure 5.** R-loops are high-affinity substrates for the Smc5/6 complex. (**A**) Coomassie blue stained gel showing the purified human SMC5/6 complex used in R/D-loop binding experiments. All the subunits of the SMC5/6 complex migrate in SDS-PAGE at the positions of the native full-length proteins. (**B**) Schematic representation of the reaction steps for the production of [$^{32}$P]-labeled R/D-loop substrates and their expected behavior in electromobility shift assays (EMSAs) on a 6% native polyacrylamide gel. Blue and pink strands represent DNA and RNA, respectively, while the asterisk indicates the $^{32}$P label introduced at the end of the RNA/DNA strand. (**C–D**) Competitive EMSA assay to evaluate the R-loop and D-loop binding specificity of human SMC5/6 complex. [$^{32}$P]-radiolabeled probe (40 nM) and 100 x molar excess Poly [d(I-C)] were confined together with gradually increasing concentrations of SMC5/6 complex. The concentrations of SMC5/6 complex used in the assays are represented by the triangles on top of the gels and correspond to the following values (in nM): 0, 6.25, 12.5, 25,

*Figure 5 continued on next page*

*Figure 5 continued*

50, and 100. Positions of unbound substrates and SMC5/6-bound R/D-loop substrates are marked by cartoon illustrations on the side of the gels.

The online version of this article includes the following figure supplement(s) for figure 5:

**Figure supplement 1.** R-loop binding affinity for the SMC5/6 complex.

**Figure supplement 2.** R-loop binding properties of the condensin holoenzyme.

D-loops. Together, our EMSA experiments indicate that the SMC5/6 complex can efficiently associate with R-loops.

The results described above prompted us to test if R-loops are universal binding substrates for SMC complexes. To test this notion, we purified yeast condensin in monomeric and multimeric forms (*Figure 5—figure supplement 2A–B*; *St Pierre et al., 2009*; *Keenholtz et al., 2017*) and conducted EMSA experiments. While multimeric condensin bound R-loop substrates with high efficiency, similar concentrations of the enzyme in monomeric form failed to bind R-loop substrates in EMSA experiments (*Figure 5—figure supplement 2C–F*). This result suggests that the ability to bind R-loops is not an intrinsic property of monomeric condensin.

## The Smc5/6 complex stimulates the degradation of R-loops by RNase H2

Next, we investigated whether binding of the SMC5/6 complex to an R-loop substrate can affect its degradation and/or stability in vitro. RNase H1 and RNase H2 are the primary enzymes responsible for the removal of R-loops in chromosomes, but recent studies suggest that RNase H2 is the only one that acts throughout the cell cycle (*Lockhart et al., 2019*; *Zimmer and Koshland, 2016*). Consistent with this, we also observed a remarkably strong genetic interaction when combining Smc5/6 complex mutations with RNase H2 enzyme inactivation (*Figure 1—figure supplement 1B*). We therefore tested the impact of the human SMC5/6 complex on the catalytic activity of purified human RNase H2 enzyme in a reconstituted R-loop degradation assay (*Figure 6A*). Similar to the EMSA, we introduced a $^{32}$P radiolabel on the RNA moiety of our R-loop substrate to allow direct visualization of RNA degradation (*Figure 6B* and *Figure 6—figure supplement 1*). We first confirmed that human RNase H2 was able to cleave the radiolabeled RNA in a time and concentration-dependent manner (*Figure 6—figure supplement 1A–B*). We also observed that the purified RNase H2 enzyme was not active on a D-loop structure, thereby demonstrating the specificity of the enzyme in our reaction conditions (*Figure 6—figure supplement 1C*). Next, we incubated increasing concentrations of the SMC5/6 complex with the R-loop substrate in the absence of RNase H2. Under these conditions, the complex did not induce degradation of the RNA moiety or otherwise affect the stability of the R-loop structure (lanes 2–6; *Figure 6C*). However, in the presence of low levels of RNase H2 enzyme, the same concentrations of SMC5/6 complex induced a major stimulation of R-loop degradation. This led to a rapid accumulation of radiolabeled product at the bottom of the gel (lanes 8–12; *Figure 6C*), reflecting the nucleolytic processing of the RNA moiety within the R-loop by RNase H2. The stimulation of RNase H2 activity by the SMC5/6 complex was concentration-dependent and evident even at the lowest concentration of the SMC5/6 complex tested in this experiment (1.25 nM, lane 8; *Figure 6C*). To test the possibility that the SMC5/6 complex modulates RNase H2 activity by direct binding, we performed an in vitro pull-down assay and found no detectable physical interaction connecting these proteins, a result further confirmed by co-immunoprecipitation assays from yeast extracts (*Figure 6—figure supplement 2A–B*). Taken together, these results indicate that the promotion of R-loop degradation by the SMC5/6 complex does not depend on this enzyme establishing a strong physical interaction with RNase H2.

## Discussion

The incorporation of RNA in chromosomal DNA represents a unique challenge for the stability of eukaryotic genomes because this modification can occur in both physiological and pathogenic conditions. Maintaining a finely balanced cycle of R-loop formation and removal in chromosomes is crucial for the overall fitness of cells because altered RNA-DNA hybrid homeostasis can result in DNA

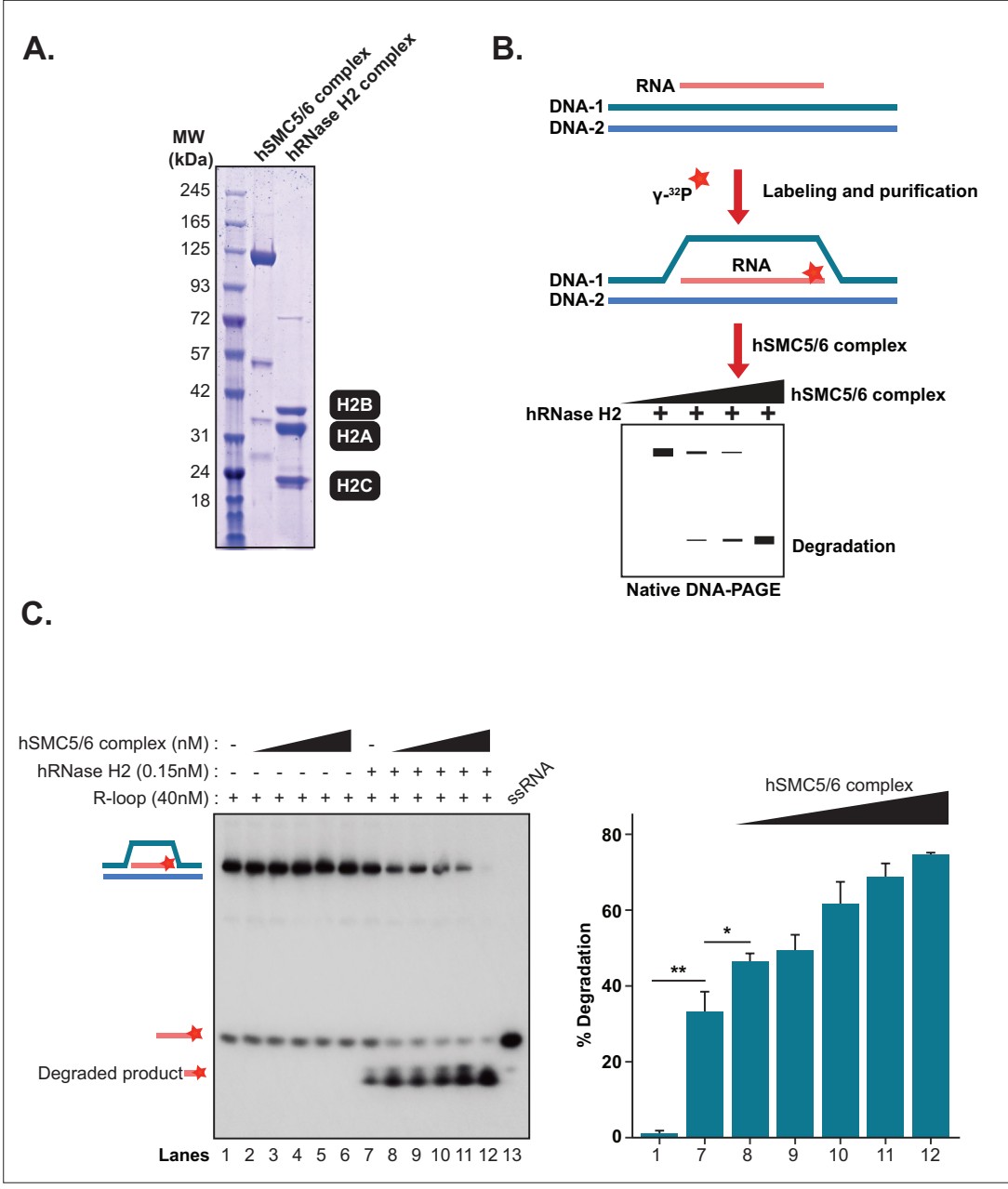

**Figure 6.** The SMC5/6 complex stimulates the degradation of R-loops by RNase H2. (**A**) Coomassie blue-stained gel showing the purity of recombinant RNase H2 (*Chon et al., 2009*) and SMC5/6 complex *Serrano et al., 2020* used in R-loop degradation assays. All the components of the SMC5/6 and RNase H2 holoenzymes migrate in SDS-PAGE at the positions expected for the native/full-length subunits of their respective complexes. (**B**) Schematic representation of the steps involved in the production of a radiolabeled R-loop probe and in the RNase H2 degradation assay. Blue and pink strands represent DNA and RNA, respectively, while the asterisk marks the $^{32}$P label introduced in the RNA strand of the R-loop structure. (**C**) R-loop degradation assay conducted in the presence of human RNase H2 (0.15 nM) and increasing concentration of human SMC5/6 complex (1.25 nM, 2.5 nM, 5 nM, 10 nM, 20 nM). The bar graph (next to the gel) shows the quantification of the degradation assay. Individual bars report the mean and SE of four independent experiments. *$p<0.05$; **$p<0.01$; ***$p<0.001$ (Student's t-test).

The online version of this article includes the following figure supplement(s) for figure 6:

**Figure supplement 1.** Biochemical properties of nucleic acid substrates and RNase H2 enzyme used in this study.

**Figure supplement 2.** Protein-protein binding experiments with human SMC5/6 complex and RNase H2 enzyme.

damage and genomic instability, ultimately contributing to the development of several pathological conditions (*Mackay et al., 2020*; *Crossley et al., 2023*). Here, we show that the Smc5/6 complex promotes the removal of toxic R-loops in eukaryotic chromosomes. While previous genetic experiments have supported a role for the Smc5/6 complex in the natural regulation of TERRA levels at telomeres (*Lafuente-Barquero et al., 2017*; *Moradi-Fard et al., 2016*), our study reports the first demonstration that Smc5/6 complex activity is essential for the removal of unscheduled R-loops from the genome. This discovery is significant because non-physiological R-loops represent the most toxic and damaging source of RNA-DNA hybrids for genome stability (*Crossley et al., 2019*; *Brambati et al., 2020*; *Petermann et al., 2022*) and failure to remove these structures exacts a heavy toll on cell fitness. Moreover, we demonstrate for the first time that the Smc5/6 complex can directly recognize R-loops, suggesting an early role in the detection and repair of these structures in vivo (see model in *Figure 7*). Consistent with this suggestion, the Smc5/6 complex has been shown by chromatin immunoprecipitation to accumulate at sites that are common R-loop enrichment zones on chromosomes, including the rDNA locus, telomeres, and highly transcribed/difficult-to-replicate chromosomal loci (*Diman et al., 2023*; *Jeppsson et al., 2023*; *Jeppsson et al., 2014*; *Pebernard et al., 2008*). We showed that inactivation of Smc5/6 components leads to an increase in R-loop formation at several of these loci in the absence of RNase H enzyme activity, a result that aligns nicely with Smc5/6 complex localization in live cells.

Our genetic enhancement results obtained with double mutants must be interpreted carefully because they involve conditional/hypomorphic alleles of Smc5/6 complex components. Synthetic or enhancement phenotypes involving non-null alleles reflect the contribution of two mutations to the same cellular process, but not necessarily or exclusively in the same molecular pathway (i.e., interactions 'within pathways' and 'between pathways;' reviewed in *Huang and Sternberg, 1995*; *Boone et al., 2007*; *Roth et al., 2009*). As such, the exacerbation of the DNA damage sensitivity of RNase H mutants by temperature-sensitive alleles of the Smc5/6 complex may be the consequence of their effects on RNA-DNA hybrid removal (i.e., thus reflecting a 'within pathway' contribution relative to RNase H1/H2) and their roles in additional biochemical pathways distinct from RNA-DNA hybrid degradation but still relevant to R-loop detoxification. We favor a model where the Smc5/6 complex acts at two distinct levels –within and between pathways– in the cellular response to R-loop formation (*Figure 7*). First, Smc5/6 contributes to RNase H-dependent removal of RNA-DNA hybrids from genomic DNA, as shown in *Figures 1 and 2*. In the execution of this function, the Smc5/6 role is substantial but not as extensive as that of RNase H enzymes (more on this below). Second, the Smc5/6 complex plays an important role in the maintenance of DNA replication fork stability, as previously established (*Peng and Zhao, 2023*). This role is crucial for cellular fitness in the presence of elevated R-loop levels because these structures often disrupt replication fork progression and can lead to fork collapse (*Crossley et al., 2019*; *Brambati et al., 2020*; *Petermann et al., 2022*; *Kemiha et al., 2021*). In this context, losing both RNase H and Smc5/6 complex activities will have consequences well beyond those observed in individual mutants because it will increase R-loop formation in a context where cells have lost the ability to cope with stress at replication forks. This 'dual hit' will render cells hypersensitive to R-loops, thus explaining the synthetic effects of combining *smc5/6* and *rnase H* mutations. Interestingly, the dual hit scenario leading to additive phenotypes appears to be the prevalent paradigm observed with effectors of RNA-DNA hybrid metabolism. For instance, past genetic interaction studies have shown that combining *sen1-1* or *sen1-3* alleles with *rnh1Δ rnh201Δ* mutations leads to synthetic lethality (*Appanah et al., 2020*). Likewise, inactivation of the THO complex induces a substantial increase in RNA-DNA hybrids in cells defective in RNase H activity (*Yang et al., 2021*), similar to our observation with *nse4-4 rnh1Δ rnh201Δ* and *smc6-9 rnh1Δ rnh201Δ* mutant strains (*Figure 2*). This pattern of synthetic enhancements when inactivating effectors of RNA-DNA hybrid metabolism is consistent with multiple mechanisms acting independently to promote the removal of toxic or unscheduled R-loops from eukaryotic genomes.

How might the Smc5/6 complex association with R-loops promote their degradation in vivo? Hints of a potential mechanism of action come from the observation that one of the enzymes responsible for R-loop degradation, RNase H1, shows little enzymatic activity under basal conditions and requires stimulation by ancillary factors, such as RPA, to achieve maximal R-loop degradation (*Nguyen et al., 2017*). It therefore seems plausible that RNase H2 might also require the assistance of a separate stimulatory factor to achieve maximal catalytic efficiency. Testing this notion in a reconstituted R-loop

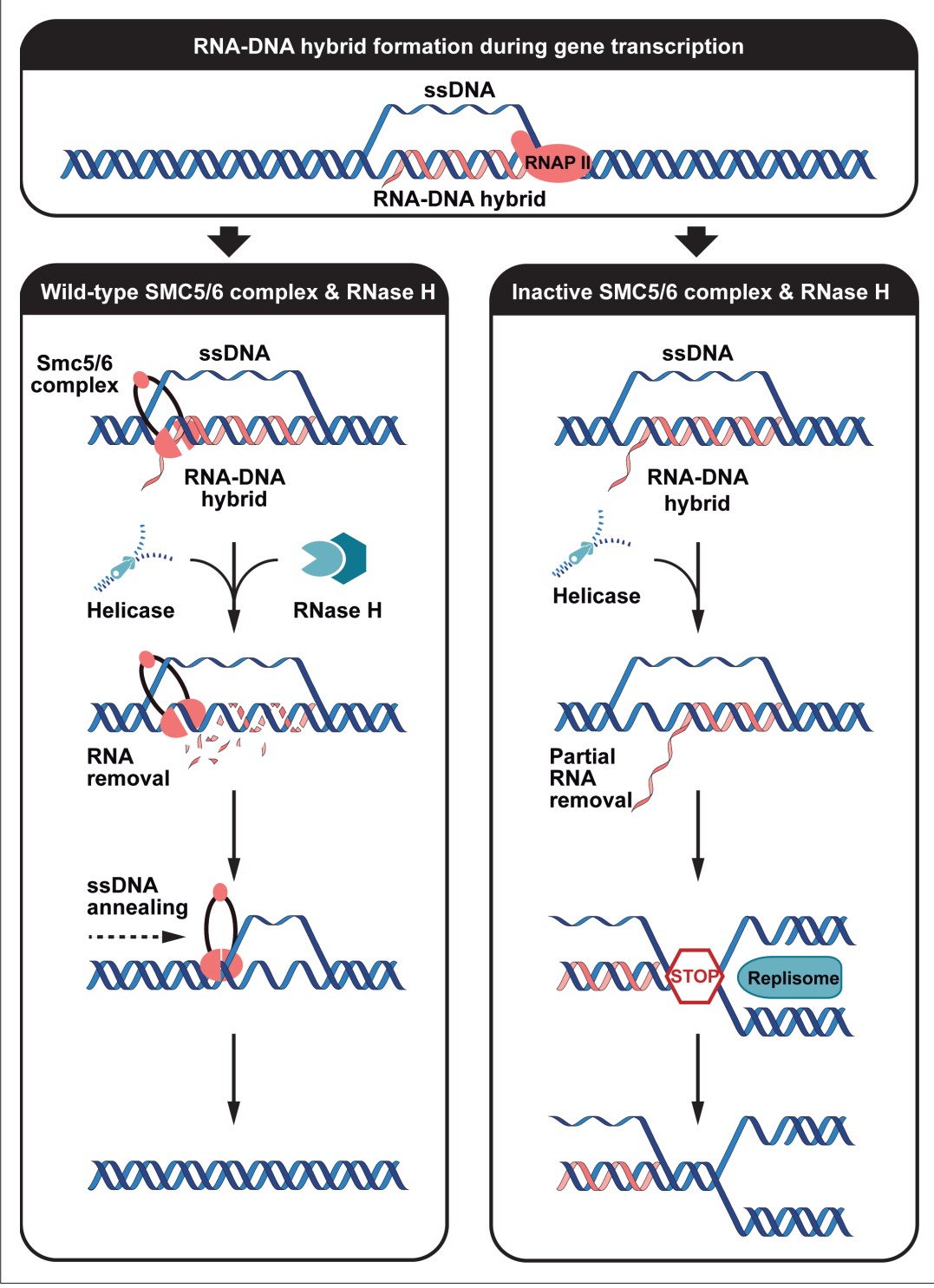

**Figure 7.** Proposed mode of action for the Smc5/6 complex during R-loop removal from chromosomes. A nascent RNA transcript synthesized during gene transcription invades separated DNA strands and forms a stable interaction with its complementary DNA stand. The Smc5/6 complex then recognizes the R-loop and associates stably with the RNA-DNA hybrid structure. RNase H2 catalytic activity is stimulated in presence of the Smc5/6 complex. Effective removal of the RNA moiety from the R-loop allows reannealing of complementary ssDNA (left panel). Based on the known DNA compaction activity of the Smc5/6 complex (*Serrano et al., 2020*), we hypothesize that this enzyme will also contribute to R-loop prevention and/or repair by facilitating reannealing of separated ssDNA formed during gene transcription and/or after removal of RNA from R-loops. Timely reannealing

*Figure 7 continued on next page*

*Figure 7 continued*
of complementary ssDNA is expected to prevent re-invasion of separated DNA strands by a new RNA transcript. In the absence of RNase H and Smc5/6 complex, the stabilized R-loop will often cause replication stress and DNA double-strand breaks (right panel). Figure prepared using Adobe Illustrator.

degradation assay confirmed the model that RNase H2 activity can be stimulated effectively, and in a dose-dependent manner by the Smc5/6 complex. The two RNase H enzymes differ, however, in that RNase H1 shows very little RNA degradation activity in the absence of RPA (*Nguyen et al., 2017*), whereas RNase H2 is moderately active as an R-loop degrading enzyme at basal state (*Figure 6* and *Figure 6—figure supplement 1*). The implication for the Smc5/6 complex is that it is probably not required to stimulate RNase H2 activity in all contexts in vivo but is likely more important in challenging environments where R-loops are highly abundant or otherwise difficult to degrade effectively by RNase H2 alone. This interpretation dovetails nicely with the synthetic interaction profiles we observed when combining Smc5/6 complex mutations with mutants that increase R-loop formation at selected genomic locations (*Figure 4*). Taken together, our genetic analyses indicate that the repertoire of genomic lesions that are substrates for RNase H2 and the Smc5/6 complex in vivo is not fully overlapping.

An RNase H2 stimulatory role for the Smc5/6 complex is compelling because it provides a cellular capacity/buffer to address substantial fluctuations in the total load of RNA-DNA hybrids produced under physiological and non-physiological conditions (*Mackay et al., 2020*; *Petermann et al., 2022*). Consequently, reducing the total load of RNA-DNA hybrids or altering its sources of origin is expected to modify the requirement for the Smc5/6 complex in R-loop metabolism (*Lafuente-Barquero et al., 2017*). How might Smc5/6 stimulate RNase H2 enzymatic activity? This is a question for a future study, but it has not escaped our attention that RNase H1 is strongly stimulated by a ssDNA binding protein, RPA (*Nguyen et al., 2017*), a biochemical property also encoded in the Smc5/6 complex (*Serrano et al., 2020*; *Roy and D'Amours, 2011*; *Roy et al., 2011*; *Roy et al., 2015*). Separate from this possibility, the Smc5/6 complex plays a vital role in promoting RPA binding and maintenance at ssDNA during homologous recombination (*Tanasie et al., 2022*), a function that could indirectly stimulate RNase H1 activity and R-loop repair (*Nguyen et al., 2017*). Addressing these possibilities will require the identification of mutations abrogating the ssDNA binding activity of the Smc5/6 complex, a difficult feat for a holoenzyme known to associate with DNA through multiple different binding modes and domains (i.e. topological and electrostatic; *Serrano et al., 2020*; *Roy et al., 2015*).

Up to now, the Smc5/6 complex has been thought of primarily as a genome stability factor associated with the repair of DSBs, recovery of stalled replication forks, telomeric length maintenance, and virus restriction (*Peng and Zhao, 2023*). While it is not evident why the Smc5/6 complex would be involved in such a diverse and loosely connected group of cellular functions, it is nevertheless clear that failure to execute these functions generates toxic recombination intermediates in vivo. Under normal circumstances, the Smc5/6 complex binds to ssDNA intermediates and branched/structured substrates at DNA repair sites and the association of the complex to these DNA intermediates provides a platform for repair factors to resolve toxic DNA lesions (*Serrano et al., 2020*; *Gutierrez-Escribano et al., 2020*; *Tanasie et al., 2022*). It is interesting to note that most of the nuclear processes outlined above involve the formation of RNA-DNA hybrids in one form or another (*Brambati et al., 2020*; *Petermann et al., 2022*). As such, the function we uncovered for the Smc5/6 complex in RNA-DNA metabolism may be a unifying role that explains its involvement in such a diverse repertoire of cellular functions. More work will be required to test this exciting possibility.

In conclusion, our work demonstrates a direct and active involvement of the Smc5/6 complex in the removal of R-loops from eukaryotic chromosomes. We showed the Smc5/6 complex binds strongly to RNA-DNA hybrid structures formed during active gene transcription and telomere length regulation, and subsequently promotes the removal of these toxic structures via the stimulation of RNase H2 enzymatic activity. This work uncovered a previously unanticipated contribution of the Smc5/6 complex in genome stability with important ramifications for the health and disease of all eukaryotic organisms.

## Methods

### Yeast strains and cell viability assay

All yeast strains used in this study are derivatives of strain K699/K700. The complete list of the relevant genotypes of the strains used in the study are provided in *Table 1*. Yeast growth conditions, procedures for genetic analysis, and creation of strains carrying relevant mutations was performed as previously described in *Serrano et al., 2020*. To create double mutant strains, haploid mutants carrying the specified alleles were mated to produce heterozygous diploid yeasts. Sporulation and dissection of diploid strains was subsequently performed at 23 °C. For the cell viability assay, performed under conditions of DNA damage or replication stress, yeast strains were grown on a solid medium containing MMS, 4-NQO, and HU at different temperatures. Specifically, a fivefold dilution series of wild-type and mutant yeast cultures (the first spot on the left side of the plate corresponds to a culture at $OD_{600}$ of 0.2) were spotted on solid YPD (yeast extract, peptone, 2% glucose) with or without the presence of DNA-damaging agent and grown in temperature-controlled incubators for 28–72 hr before scanning the plates in a scanner (*Serrano et al., 2020*). Strains expressing RNase H1 were generated by integrating *YIplac204::Pgal1::RNH1* at the *TRP1* locus and integration was confirmed by PCR screening. To create strains expressing the AID enzyme, we transformed relevant strains tagged with GFP at the C-terminus of Rad52 with *pESC-LEU-HsAIDSc* plasmid (Addgene plasmid #60810) (*Mayorov et al., 2005*). A complete list of plasmids used in this study is also provided in *Table 2*.

### Chromatin spread immunolabeling with the S9.6 antibody

Chromatin spreads were performed as described previously (*Grubb et al., 2015*) with some minor modifications. Briefly, cells from the mid-log phase grown in YPD at 23 °C were collected and washed in 1 mL ZK buffer (25 mM Tris, pH 7.5, 0.8 M KCl) and were resuspended in ZK buffer supplemented with 1 M DTT. Spheroplasting of cells was achieved by the addition of 5 µl of Zymolyase (20 mg/ml) and incubation at 30 °C with gentle rotation. Subsequently, spheroplast cells were centrifuged (2000 rpm/5 min) and resuspended in MES/Sorbitol buffer (0.1 M MES pH 6.5, 0.5 mM $MgCl_2$, 1 mM EDTA, 1 M sorbitol). Next, cells were added to the glass slide (Corning) and immediately were fixed and lysed by the addition of a fixative solution (3% paraformaldehyde in 4% sucrose) and 1% NP-40 substitute solution. Cells were spread using a plastic pipette rolled from one end of the slide to the other end. Slides with chromatin spreads were dried overnight. The next day, slides were washed with 1 x TBS (Tris-buffered saline) for 10 min and blocked for 15 min with 5% BSA (Bovine serum albumin) in 1 x TBS. Chromatin spreads were incubated with mouse monoclonal antibody S9.6 (MABE1095) (1:250 dilution) for overnight followed by Cy3-conjugated goat anti-mouse antibody (Jackson Laboratories, #115-165-003) (1:700 dilution) for 2 hr. Nuclei were counterstained with 50 µl of VectaShield (Vector Laboratories, CA) plus 1 x DAPI (4',6- diamidino-2-phenylindole) and sealed with nail polish. Images were acquired using Nikon Eclipse Ti2 inverted microscopy with an oil immersion 100 x objective. For each replicate (n>3), about 150 nuclei were visualized and manually counted to obtain the fraction with detectable RNA-DNA hybrid foci using the 3D measurement module of the NIS-Elements software (Nikon Instruments Inc). Fluorescence intensity of RNA-DNA hybrid structures (arbitrary units; A.U.) was quantified using ImageJ (N.I.H, USA).

### RNA-DNA hybrid immunoprecipitation followed by qPCR

Mid-log cultures grown in YPD at 23 °C were collected. RNA-DNA hybrids were processed and analyzed as described in *El Hage and Tollervey, 2018*. Real-time quantitative PCR was performed at the indicated regions using the SsoAdvanced SYBR Green PCR Master Mix (Bio-Rad) with a CFX384 Real-Time PCR System (C-1000 Touch Thermal Cycler). Data was analyzed using the CFX Maestro Bio-Rad software and the relative abundance of RNA-DNA hybrid immunoprecipitated in each region was normalized to the signal obtained in the inputs. Average and standard error of at least three independent experiments are shown.

### Purification of the SMC5/6 complex

The human SMC5/6 core complex was purified using a triple affinity purification approach followed by size exclusion chromatography, as described by *Serrano et al., 2020* with minor modifications. 35 L of an *S. cerevisiae* strain overexpressing the core complex was cultured under optimal growth conditions in a bioreactor (Techfors-S-42L) to an $OD_{600}$ of 0.7–1.0. Protein expression was induced by

**Table 1.** Yeast strains used in this study.

| Figure | Strain name | Relevant genotype details |
|---|---|---|
| *Figure 1* | D7528 | *MATa* |
| | D6531 | *MATa rnh1::HIS3MX6 rnh201::kanMX6* |
| | D6533 | *MATa rnh1::HIS3MX6 rnh202::kanMX6* |
| | D6535 | *MATa rnh1::HIS3MX6 rnh203::kanMX6* |
| | D7799 | *MATa nse4-4::URA3* |
| | D7055 | *MATa nse4-4::URA3 rnh1::HIS3MX6 rnh201::kanMX6* |
| | D7057 | *MATa nse4-4::URA3 rnh1::HIS3MX6 rnh202::kanMX6* |
| | D7084 | *MATa nse4-4::URA3 rnh1::HIS3MX6 rnh203::kanMX6* |
| | D7795 | *MATa smc6-9::NAT* |
| | D7159 | *MATa smc6-9::NAT rnh1::HIS3MX6 rnh201::kanMX6* |
| | D7357 | *MATa smc6-9::NAT rnh1::HIS3MX6 rnh202::kanMX6* |
| | D7157 | *MATa smc6-9::NAT rnh1::HIS3MX6 rnh203::kanMX6* |
| *Figure 2A* | D7528 | *MATa* |
| | D6531 | *MATa rnh1::HIS3MX6 rnh201::kanMX6* |
| | D6533 | *MATa rnh1::HIS3MX6 rnh202::kanMX6* |
| | D6535 | *MATa rnh1::HIS3MX6 rnh203::kanMX6* |
| | D7799 | *MATa nse4-4::URA3* |
| | D7055 | *MATa nse4-4::URA3 rnh1::HIS3MX6 rnh201::kanMX6* |
| | D7057 | *MATa nse4-4::URA3 rnh1::HIS3MX6 rnh202::kanMX6* |
| | D7084 | *MATa nse4-4::URA3 rnh1::HIS3MX6 rnh203::kanMX6* |
| | D7795 | *MATa smc6-9::NAT* |
| | D7159 | *MATa smc6-9::NAT rnh1::HIS3MX6 rnh201::kanMX6* |
| | D7357 | *MATa smc6-9::NAT rnh1::HIS3MX6 rnh202::kanMX6* |
| | D7157 | *MATa smc6-9::NAT rnh1::HIS3MX6 rnh203::kanMX6* |
| *Figure 2B* | D8122 | *MATa trp1-1::Pgal1::TRP1* |
| | D8120 | *MATa trp1-1::Pgal1::RNH1::TRP1* |
| | D8177 | *MATa rnh1::HIS3MX6 rnh201::kanMX6 trp1-1::Pgal1::TRP1* |
| | D8407 | *MATa rnh1::HIS3MX6 rnh201::kanMX6 trp1-1::Pgal1::RNH1::TRP1* |
| | D8128 | *MATa smc6-9::NAT rnh1::HIS3MX6 rnh201::kanMX6 trp1-1::Pgal1::TRP1* |
| | D8119 | *MATa smc6-9::NAT rnh1::HIS3MX6 rnh201::kanMX6 trp1-1::Pgal1:: RNH1::TRP1* |
| | D8130 | *MATa nse4-4::URA3 rnh1::HIS3MX6 rnh203::kanMX6 trp1-1::Pgal1::TRP1* |
| | D8221 | *MATa nse4-4::URA3 rnh1::HIS3MX6 rnh203::kanMX6 trp1-1::Pgal1:: RNH1::TRP1* |
| *Figure 3A* | D7528 | *MATa* |
| | D6535 | *MATa rnh1::HIS3MX6 rnh203::kanMX6* |
| | D7084 | *MATa nse4-4::URA3 rnh1::HIS3MX6 rnh203::kanMX6* |

*Table 1 continued on next page*

*Table 1 continued*

| Figure | Strain name | Relevant genotype details |
|---|---|---|
| *Figure 3B* | D8852 | *MATa rnh1::HIS3MX6 rnh201::kanMX6 RAD52=EGFP::KanMX6 [p6-YCplac111]* |
| | D8849 | *MATa rnh1::HIS3MX6 rnh201::kanMX6 nse4-4::URA3 RAD52=EGFP::KanMX6 [p6-YCplac111]* |
| | D8611 | *MATa rnh1::HIS3MX6 rnh201::kanMX6 RAD52=EGFP::KanMX6 [p1895-pESC-LEU-HsAIDSc]* |
| | D8613 | *MATa rnh1::HIS3MX6 rnh201::kanMX6 nse4-4::URA3 RAD52=EGFP::KanMX6 [p1895-pESC-LEU-HsAIDSc]* |
| *Figure 4, Figure 4—figure supplement 1* | D8457 | *MATa hpr1::HIS3MX6* |
| | D8534 | *MATa smc6-9::NAT hpr1::HIS3MX6* |
| | D8368 | *MATa sen1-1::Tadh1::HIS3MX6* |
| | D8376 | *MATa sen1-1::Tadh1::HIS3MX6 nse4-4::URA3* |
| | D8438 | *MATa sen1-1::Tadh1::HIS3MX6 smc6-9::NAT* |
| | D8573 | *MATa sen1-3[R1605K]::Tadh1::HIS3MX6* |
| | D8603 | *MATa sen1-3[R1605K]::Tadh1::HIS3MX6 nse4-4::URA3* |
| | D8601 | *MATa sen1-3[R1605K]::Tadh1::HIS3MX6 smc6-9::NAT* |
| | D6537 | *MATa pol2[M644G]* |
| | D8645 | *MATa pol2[M644G] nse4-4::URA3* |
| | D8643 | *MATa pol2[M644G] smc6-9::NAT* |
| | D8691 | *MATa rat1-1* |
| | D8789 | *MATa rat1-1 nse4-4::URA3* |
| | D8783 | *MATa rat1-1 smc6-9::NAT* |
| | D8844 | MATa rat1-1 sen1-1::Tadh1::HIS3MX6 smc6-9::NAT |
| *Figure 1—figure supplement 1A* | D7528 | *MATa* |
| | D6531 | *MATa rnh1::HIS3MX6 rnh201::kanMX6* |
| | D6533 | *MATa rnh1::HIS3MX6 rnh202::kanMX6* |
| | D6535 | *MATa rnh1::HIS3MX6 rnh203::kanMX6* |
| | D7799 | *MATa nse4-4::URA3* |
| | D7055 | *MATa nse4-4::URA3 rnh1::HIS3MX6 rnh201::kanMX6* |
| | D7057 | *MATa nse4-4::URA3 rnh1::HIS3MX6 rnh202::kanMX6* |
| | D7084 | *MATa nse4-4::URA3 rnh1::HIS3MX6 rnh203::kanMX6* |
| | D7795 | *MATa smc6-9::NAT* |
| | D7159 | *MATa smc6-9::NAT rnh1::HIS3MX6 rnh201::kanMX6* |
| | D7357 | *MATa smc6-9::NAT rnh1::HIS3MX6 rnh202::kanMX6* |
| | D7157 | *MATa smc6-9::NAT rnh1::HIS3MX6 rnh203::kanMX6* |

*Table 1 continued*

| Figure | Strain name | Relevant genotype details |
|--------|-------------|---------------------------|
| Figure 1—figure supplement 1B | D7528 | *MATa* |
| | D9636 | *MATa rnh1::HIS3MX6* |
| | D9613 | *MATa nse4-4::URA3 rnh1::HIS3MX6* |
| | D9607 | *MATa smc6-9::NAT rnh1::HIS3MX6* |
| | D9637 | *MATa rnh201::kanMX6* |
| | D9638 | *MATa rnh202::kanMX6* |
| | D9639 | *MATa rnh203::kanMX6* |
| | D9615 | *MATa nse4-4::URA3 rnh201::kanMX6* |
| | D9617 | *MATa nse4-4::URA3 rnh202::kanMX6* |
| | D9620 | *MATa nse4-4::URA3 rnh203::kanMX6* |
| | D9605 | *MATa smc6-9::NAT rnh201::kanMX6* |
| | D9609 | *MATa smc6-9::NAT rnh202::kanMX6* |
| | D9612 | *MATa smc6-9::NAT rnh203::kanMX6* |
| Figure 1—figure supplement 1C | D7528 | *MATa* |
| | D6531 | MATa rnh1::HIS3MX6 rnh201::kanMX6 |
| | D6533 | MATa rnh1::HIS3MX6 rnh202::kanMX6 |
| | D6535 | MATa rnh1::HIS3MX6 rnh203::kanMX6 |
| | D8368 | *MATa sen1-1::Tadh1::HIS3MX6* |
| | D7671 | *MATa rad5-535 mms21-H202Y::URAMX6* |
| | D9633 | *MATa rnh1::HIS3 rnh201::kanMX6 mms21-H202Y::URAMX6* |
| | D9632 | *MATa rnh1::HIS3 rnh202::kanMX6 mms21-H202Y::URAMX6* |
| | D9650 | *MATa rnh1::HIS3 rnh203::kanMX6 mms21-H202Y::URAMX6* |
| | D9634 | *MATa sen1-1::Tadh1::HIS3MX6 mms21-H202Y::URAMX6* |
| Figure 3—figure supplement 1 | D9105 | *MATa RAD52=EGFP::KanMX6 [p6-YCplac111]* |
| | D9101 | *MATa nse4-4::URA3 RAD52=EGFP::KanMX6 [p6-YCplac111]* |
| | D9107 | *MATa RAD52=EGFP::KanMX6 [p1895-pESC-LEU-HsAIDSc]* |
| | D9105 | *MATa nse4-4::URA3 RAD52=EGFP::KanMX6 [p1895-pESC-LEU-HsAIDSc]* |
| Figure 6—figure supplement 2 | D9272 | *MATa rad5-535 SMC5=3xSTII::TRP1* |
| | D9274 | *MATa rad5-535 SMC5=3xSTII::TRP1 RNH201=13xMYC::kanMX6* |
| | D9585 | *MATa rad5-535 SMC5=3xSTI::TRP1 RNH202=13xMYC::kanMX6* |
| | D9264 | *MATa rad5-535 SMC5=3xSTII::TRP1 NSE3=13MYC::HIS3MX6* |

**Table 2.** Plasmids used in this study.

| Figure | Plasmid number | Relevant details |
|--------|----------------|------------------|
| Figure 2 | p32 | *YIplac204/GAL1-10* |
| | p355 | *YIplac204_Pgal1_RNH1* |
| Figure 3 | p6 | *YCplac111* |
| | p1895 | *pESC-LEU-HsAIDSc* |
| Figure 6 | p1906 | *pET-hH2ABC* |

the addition of 2% galactose and cells were grown further for 16 hr at 18 °C. Briefly, yeast pellets were resuspended in 200 ml buffer N (50 mM $K_2HPO_4$ / $KH_2PO_4$ pH 8, 50 mM Tris-HCl pH 8.0, 500 mM NaCl, 10% glycerol, 0.5% Triton X-100, 2 mM β-Mercaptoethanol) supplemented with 20 mM imidazole and protease inhibitors (E64, Pepstatin A, 4-(2-aminoethyl) benzene sulfonyl fluoride hydrochloride [AESBF]). Yeast popcorn is made by the dropwise freezing of the cell suspension in liquid nitrogen. The popcorns were further lysed two cycles in a freezer mill. The lysates were resuspended in 1 L of buffer N and passed through a high-pressure homogenizer (Avestin EmulsiFlex-C3) at an operating pressure of 25,000 psi. The final lysate was centrifuged at 24,000 rpm for 45 min at 4 °C. The soluble lysates were passed through a column packed with Nickel-NTA resin at a flow rate of 5 ml/min for slower binding. The unbound fractions from the first purification column were loaded to a second Ni-NTA column for a second-round binding to maximise yield of purified protein. Both the columns were washed with 10 column volumes (CV) of buffer N supplemented with 60 mM imidazole. Complex was eluted with buffer SB (50 mM Tris-HCl pH 8.0, 500 mM NaCl, 10% glycerol, 0.5% Tween 20, 2 mM βME) supplemented with 500 mM imidazole. The combined fractions from both the columns were loaded into a StrepTractin XT 5 mL column using an AKTA prime FPLC purification system. The column was programed to wash with 10 CV of buffer SB supplemented with 0.5% Triton X-100 and eluted with 5 CV of buffer GB (25 mM $K_2HPO_4$/$KH_2PO_4$ pH 8, 500 mM NaCl, 10% glycerol, and 2 mM βME) supplemented with 50 mM biotin. The elution was mixed and incubated with 5 ml of pre-equilibrated Glutathione S-transferase (GST)-Sepharose resin, in GST binding buffer GB (20 mM $K_2HPO_4$/$KH_2PO_4$ pH 8, 200 mM NaCl, 10% glycerol and 2 mM DTT) for 2 hr at 4 °C. The resin was washed with 10 CV of buffer GEB (50 mM Tris-HCl pH 8.0, 500 mM NaCl, 10% glycerol, and 2 mM βME). The SMC5/6 complex was eluted with 5 CV of buffer GEB supplemented with 25 mM of reduced Glutathione. Linker, poly-histidine, Strep-tag II, and GST tags were cleaved by an overnight digestion with 1 mg of TEV protease per 4 mg/mL of fusion protein. Digestion was carried out in GEB buffer supplemented with 1 mM DTT. Digestion product was loaded into a Superose 6 10/300 size exclusion chromatography column in GF buffer (50 mM Tris-HCl pH 8.0, 500 mM NaCl, 10% glycerol, and 2 mM βME) in order to remove the cleaved tags, digested linker, and TEV protease. Elution fractions containing highly purified and stoichiometric complexes were concentrated, quantified, snap-frozen, and stored at −80 °C.

## Purification of active RNase H2 enzyme complex

A polycistronic vector allowing the co-expression of all the subunits of human RNase H2 (*pET-hH2ABC);* was obtained from Robert J. Crouch (NIH). All the subunits of the holoenzyme–namely, RNase H2A, H2B, and H2C–are expressed from this vector as N-terminal hexahistidine fusion proteins. For purification, *Escherichia coli* BL21 was transformed with *pET-hH2ABC,* and 6 L of culture was grown at 37 °C to an $OD_{600}$ of 0.4–0.6 before being induced with 0.3 mM of IPTG. The culture was grown further at 18 °C for 16 hr after induction. Bacterial pellets were resuspended in 100 ml buffer N (50 mM $K_2HPO_4$ /$KH_2PO_4$ pH 8, 50 mM Tris-HCl pH 8.0, 500 mM NaCl, 10% glycerol, 0.5% Triton X-100, 2 mM βME) supplemented with 20 mM imidazole and protease inhibitors (E64, Pepstatin A, AESBF). The lysate was passed twice through a high-pressure homogenizer (Avestin EmulsiFlex-C3) at an operating pressure of 15,000 psi and centrifuged at 24,000 rpm for 45 min at 4 °C. The soluble lysates were passed through a column packed with Ni-NTA resin at a flow rate of 5 ml/min for slower binding. The column was washed with 10 column volumes (CV) of buffer N supplemented with 60 mM imidazole. Complex was eluted with buffer SB (50 mM Tris-HCl pH 8.0, 500 mM NaCl, 10% glycerol, 0.5% tween 20, 2 mM βME) supplemented with 500 mM imidazole. The eluted fractions were dialyzed against 20 mM HEPES pH 7.6, 250 mM NaCl, 10% glycerol, and 1 mM DTT, quantified, snap frozen, and stored at −80 °C.

## RNA-DNA hybrid probe synthesis

The DNA or RNA oligos are 5′-labeled with ATP-[γ$^{32}$P] (PerkinElmer Life Sciences) using T4 polynucleotide kinase (New England BioLabs). Radiolabeled oligos were then annealed to a complementary strand by heating to 95 °C and slow cooling over a long period of time in PNK buffer (70 mM Tris-HCl pH 7.6, 10 mM $MgCl_2$, 5 mM DTT). Annealed substrates were separated from free ATP-[γ$^{32}$P] on an 8% native PAGE in Tris Borate/EDTA buffer at room temperature. The gel band corresponding to the

annealed substrate was excised, purified, and finally eluted. The eluted substrates were quantified (nM) using a scintillation counter.

Two types of R-loop substrates were synthesized. First, the R-loop substrate was constructed by annealing [$^{32}$P]-labeled RNA strand (DD 4265) with DNA strand 1 (DD 4263) and DNA strand 2 (DD 4264). Another R-loop substrate was constructed by annealing [$^{32}$P]-labeled DNA strand 2 (DD 4264) with DNA strand 1 (DD 4263) and RNA strand (DD 4265). Second, a D-loop substrate was also generated by annealing [$^{32}$P]-labelled DNA strand 2 with DNA strand 1 and a DNA strand 3 (DD 4266). A single [$^{32}$P]-labeled ssDNA strand (DD4264) was used as a control for electrophoretic migration in a gel.

The sequence of oligonucleotides used as in vitro substrates are: DNA strand 1 (DD 4263; 1.5 μg/μl or 100 μM): 5'GGGTGAACCTGCAGGTGGGCGGCTGCTCATCGTAGGTTAGTTGGTAGAATTCGGCA GCGTC-3' (61 mer); DNA strand 2 (DD 4264; 1.8 μg/μl or 100 μM): 5'GACGCTGCCGAATTCTACCA GTGCCTTGC TAGGACATCTTTGCCCACCTGCAGGTTCACCC-3' (61 mer); RNA strand 1 (DD 4265; 0.5 μg/μl or 100 μM): 5'-AAAGArUGrUCCrUAGCAAGGCAC-3' (21 mer); DNA strand 3 (DD 4266; 0.6 μg/μl or 100 μM): 5'-AAAGATGTCCTAGCAAGGCAC-3' (21 mer).

## In vitro R-loop and D-loop binding assays

The DNA binding activity of human SMC5/6 complex and yeast condensin was determined by electrophoretic mobility shift assay (EMSA). The condensin holoenzyme was purified according to a published procedure (*St Pierre et al., 2009*). Reactions containing 40 nM [$^{32}$P]-labeled oligonucleotides and the indicated concentrations of SMC5/6 complex and condensin (i.e. 0, 6.25, 12.5, 25, 50, and 100 nM) were incubated in binding buffer A (25 mM MOPS [morpholinepropanesulfonic acid] pH 7.6, 60 mM KCl, 0.2% NP40, 2 mM DTT, 5 mM MgCl$_2$) in a total volume of 15 μl. Reactions were incubated at 24 °C for 10 min and loaded on a 6% acrylamide gel, electrophoresed at 150 volts for 240 min in 1 X TBE buffer for EMSA. Gels were then dried onto DE81 filter paper and visualized by autoradiography. To evaluate the specificity of the SMC5/6 complex binding to R-loop as compared to D-loop, increasing concentrations of human SMC5/6 complex (i.e. 0, 6.25, 12.5, 25, 50, and 100 nM) were incubated with 100 x molar excess concentration of unlabeled poly-deoxy-inosinic-deoxy-cytidylic acid (poly[d(I-C)], Roche, Cat. No. 0108812001) along with 40 nM of [$^{32}$P]-labelled R-loop and D-loop, respectively, followed by EMSA. Quantification of unbound and protein-bound DNA was performed with ImageJ using the histogram function. Data was fitted to a Hill equation with dissociation constants ($K_D$) using GraphPad Prism 9.0 (GraphPad Software Inc).

## R-loop RNase H2 assay

RNase assays were performed in Buffer C (20 mM HEPES [4-(2-hydroxyethyl)–1-piperazineethane sulfonic acid] pH 7.5, 150 mM NaCl, 10 mM MgCl$_2$, 0.5 mM DTT). The R-loop (40 nM) substrates were pre-incubated with SMC5/6 complex at the indicated concentration in buffer C for 10 min at 24 °C followed by the addition of RNase H2 complex for 7 min at the same reaction conditions. Reactions were deproteinized in a one-fifth volume of stop buffer (Buffer A, 1% SDS, 5 mM EDTA, and 0.2 mg/ml proteinase K) for 15 min at 24 °C. Reactions were loaded on an 8% acrylamide gel, electrophoresed at 150 volts for 150 min, dried onto DE81 filter paper, and visualized by autoradiography.

## AID-induced Rad52 foci assay by fluorescence microscopy

Detection of R-loops through AID-induced DNA damage and subsequent rise of Rad52 foci was performed as described in *Cañas et al., 2022*. Strains were grown at 23 °C in minimal media followed by galactose induction for 2 hr to overexpress AID enzymes and collected. Cells were fixed in formaldehyde (10% in 0.1 M KPO$_4$ pH 6.4) for 30 min at room temperature and washed twice in 0.1 M KPO$_4$ pH 7.0 buffer. Staining of the nuclei was performed with DAPI at a final concentration of 2 μg/ml in cells suspended in 0.1 M KPO$_4$ pH 7.0 buffer. For DAPI staining and Rad52-GFP visualization, images were acquired using Nikon Eclipse Ti2 inverted microscopy with an oil immersion 100 x objective. For each replicate (n=3), about 100 cells were visualized and manually counted to obtain the fraction with detectable Rad52 foci using the 3D measurement module of the NIS-Elements software (Nikon Instruments Inc).

## In vitro RNase H2 and SMC5/6 complex binding assay

Active RNase H2 enzyme and SMC5/6 complex were purified as described above. The resulting complexes carry tandem Strep-tag II (3xSTII) tags on SMC6 and a 10xHis tag on NSMCE4–SMC5. RNase H2 subunits were tagged with a 6xHis tag. For the pull-down procedure, Streptactin XT resin (10 μL per reaction) was washed with ddH$_2$O and with binding buffer (50 mM HEPES-NaOH [4-(2-hydroxyethyl) piperazine-1-ethanesulfonic acid, N-(2-hydroxyethyl)piperazine-N′-(2-ethanesulfonic acid)–sodium hydroxide] pH 7.5, 150 mM NaCl, 5% glycerol, 1% BSA, 5 mM MgCl$_2$) before being incubated in binding buffer in a total volume of 300 μL for 2 hr at 4 °C with or without 0.8 μM human SMC5/6 complex. The resins were then washed with wash buffer (50 mM HEPES-NaOH pH 7.5, 150 mM NaCl, 5% glycerol, 0.2% NP40, 5 mM MgCl$_2$). For each condition, 10 μL of Streptactin XT resin with or without 1.95 μM of human RNase H2 were incubated in a total volume of 300 μL of binding buffer at 4 °C overnight (~12 hr). The reactions were then washed with wash buffer. The pellet was resuspended in wash buffer and 4 x sample buffer (90% 4 x Laemmli sample buffer, 10% β-mercaptoethanol) at a ratio of 1:1. The reactions were then loaded on an 4–12% Bis-Tris SDS-PAGE gel and ran in 1 x MOPS buffer for 2 hr at 120 Volts, before being analyzed by immunoblotting with 1:2500 dilution of anti-HIS antibody (Qiagen 34660).

## In vivo co-immunoprecipitation assay (co-IP)

Co-immunoprecipitation assay was performed as described previously (*Pastic et al., 2024*) with minor modifications. Yeast cultures (50 mL) were grown in YPD at 23 °C to an OD$_{600}$ of 0.7–0.8 and washed with ddH$_2$O before being snap-frozen in liquid nitrogen and stored at –80 °C. For the co-IP method, magnetic protein G beads (Dynabeads Protein G; Thermo Fisher Scientific, Cat #1003D) were washed with the PBS-BSA buffer (1 x PBS [Phosphate buffered saline], 1% BSA) and incubated for 2 hr at 4 °C with 4 μg of mouse anti-Strep antibody (Qiagen 34850) antibody or 4 μg of mouse IgG Isotype control (Invitrogen 02–6502) per strain to confirm the specificity of the anti-Strep antibody. Cells were lysed by vigorous shaking with glass beads in 800 μL of co-IP lysis buffer (50 mM HEPES–KOH pH 7.5, 140 mM NaCl, 1% Trition X-100, 0.1% Sodium deoxycholate, 1 mM EDTA pH 8.0, 1 mM AEBSF, 10 μM Pep-A, 10 μM E64). The lysates (~1 mL) were then cleared by centrifugation. Once the beads were washed with the PBS-BSA buffer, the lysate were equally split between the two beads conditions and incubated at 4 °C overnight (~12 hr). The beads were then washed with the co-IP lysis buffer, with high salt buffer (50 mM HEPES – KOH pH 7.5, 360 mM NaCl, 1% Trition X-100, 0.1% Sodium deoxycholate, 1 mM EDTA pH 8.0) and with TE buffer (10 mM Tris-HCl pH 8.0, 0.1 mM EDTA) and resuspended in 2 X sample buffer (4 X Sample buffer, 0.2% of BPB [bromophenol blue],~5% ddH$_2$O, 5% of β-mercaptoethanol). Proteins were separated on an SDS-PAGE gel [4–12% Bis-Tris] in 1 x MOPS buffer and ran for 2 hr at 120 Volts. Proteins were subsequently transferred on a membrane before being analyzed by immunoblotting with 1:2000 dilution of mouse anti-Strep antibody and 1:5000 dilution of mouse anti-Myc (9E10) antibody (GeneTex 369661).

## Statistical analyses

Results presented in this study are representative examples of at least three independent experiments. All statistical analyses were performed using GraphPad Prism 7 (GraphPad Software Inc). Sample size (n) and statistical tests performed in each experiment are described in the relevant figure legends. In all cases, p-values expressed as *p<0.05, **p<0.005, and ***p<0.0005 are considered significant.

## Acknowledgements

We thank Drs. Sarah Kinkley, Malika Saint, Jordi Torres-Rosell, and members of the D'Amours laboratory for their comments on the manuscript. We also thank Marco Muzi-Falconi (Università degli Studi di Milano) for providing us with RNase H mutant strains, Pascal Chartrand (Université de Montréal) for sharing the *rat1-1* mutant, and Robert J Crouch (NIH) for providing the plasmids used for the purification of RNase H2.

## Additional information

### Funding

| Funder | Grant reference number | Author |
|---|---|---|
| Canadian Institutes of Health Research | FDN-167265 | Damien D'Amours |
| Canada Research Chair in Chromatin Dynamics & Genome Architecture | CRC-2017-00064 | Damien D'Amours |

The funders had no role in study design, data collection and interpretation, or the decision to submit the work for publication.

### Author contributions

Shamayita Roy, Conceptualization, Formal analysis, Validation, Investigation, Writing - original draft, Writing – review and editing; Hemanta Adhikary, Formal analysis, Validation, Investigation, Writing - original draft, Writing – review and editing; Sarah Isler, Formal analysis, Validation, Investigation, Writing – review and editing, Sarah Isler has been added as an author at the revised stage. She has been added at this stage to perform some experiments suggested by the reviewers. We confirm that all authors agree with their inclusion and place in the author list; Damien D'Amours, Conceptualization, Supervision, Funding acquisition, Writing – review and editing

### Author ORCIDs

Shamayita Roy  https://orcid.org/0009-0001-9732-1446
Sarah Isler  http://orcid.org/0009-0005-2470-2327
Damien D'Amours  https://orcid.org/0000-0002-2183-9951

### Decision letter and Author response

Decision letter https://doi.org/10.7554/eLife.96626.sa1
Author response https://doi.org/10.7554/eLife.96626.sa2

## Additional files

### Supplementary files

MDAR checklist

### Data availability

All source Data files have been provided for Figures 2-3, Figure 5-6, and Figure supplements on the Dryad open repository site at https://doi.org/10.5061/dryad.3xsj3txpg.

The following dataset was generated:

| Author(s) | Year | Dataset title | Dataset URL | Database and Identifier |
|---|---|---|---|---|
| D'Amours D | 2024 | Data from: The Smc5/6 complex counteracts R-loop formation at highly transcribed genes in cooperation with RNase H2 | https://doi.org/10.5061/dryad.3xsj3txpg | Dryad Digital Repository, 10.5061/dryad.3xsj3txpg |

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
