## [Editor Report]

This study presents an important finding showcasing the role of Smc5-6 complex in counteracting R-loops at transcriptionally active sites. The evidence supporting the claims of the authors is solid, although inclusion of a genome-wide R-loop detection assay would have strengthened the study. The work will be of interest to scientists studying genome structure and stability.

---

## [Decision Letter]

**Decision letter after peer review:**

Thank you for submitting your article "The Smc5/6 complex counteracts R-loop formation at highly transcribed genes in cooperation with RNase H2" for consideration by *eLife*. Your article has been reviewed by 3 peer reviewers, and the evaluation has been overseen by a Reviewing Editor and Adèle Marston as the Senior Editor.

Essential revisions (for the authors):

Please address the following essential revisions:

i) Please provide genetic interaction studies of RNH1 and RNH2 (single mutants) with nse4-4 and smc6-9.

ii) For quantitation of S9.6 immunofluorescence (Figure 2A), it may serve the authors better to measure A.U. per nucleus spread rather than foci count (>10), since some of those signals appear to be clustered. Please also address how background foci are dealt with for this assay.

iii) Please provide S9.6 immunofluorescence experiments for interaction with THO complex and Sen1 (Figure 4).

iv) Please provide quantitative (dissociation constant) measurements for EMSA experiments. Competitive titration experiments with D-loop will strengthen the claim.

v) Please include key primary references in the manuscript, where missing.

*Reviewer #1 (Recommendations for the authors):*

The Authors characterize the combined impact of the Smc5/6 complex and RNase H in R-loop degradation. The work is solid and will provide an advance, but how Smc5/6 is functioning mechanistically remains obscure. The work identifies altered toxic hybrids when the smc5/6 complex is compromised however, it is difficult to distinguish between cause and effect. For example, is the in vivo localization of the Smc5/6 complex to regions of the genome altered by RNA-DNA hybrid levels?

The Authors characterize the combined impact of the Smc5/6 complex and RNase H in R-loop degradation. The work is solid and will provide an advance, but how Smc5/6 is functioning mechanistically remains obscure. Is the in vivo localization of the Smc5/6 complex to regions of the genome altered by RNA-DNA hybrid levels. What is the cause vs. what is the effect?

Please address the following:

Location of rnh201D and rnh203D are unclear from Figure 1A. I believe one is mislabelled.

Has it been determined what rnh1D alone looks like with the smc6-9 and nse4-4?

The authors state that the presence of S9.6 foci was highest in the nse4-4 rnh1Δ rnh201Δ and smc6-9 rnh1Δ rnh201Δ strains, consistent with the fact that rnh201Δ represents the deletion of the catalytic subunit of RNase H2 (Figure 2A). – How would this be interpreted in the context of rnh1Δ rnh201Δ double mutants alone, which show less hybrid formation than counterparts. Please consider rephrasing.

Importantly, overexpression of RNase H1 largely suppressed the S9.6 signal on chromatin spreads (Figure 2B), indicating that the foci described above are reflective of RNA-DNA hybrid formation in double mutant strains. (p.6) – add more detail into how this was performed in the Results section – ectopic expression?

The quantification of nse4-4 rnh1D rnh203D +RNH1 doesn't look representative of the microscopy. Also, please address why there more foci in nse4-4 rnh1D rnh203D+RNH1 compared to nse4-4 single mutants, shouldn't these be the same? Perhaps it is the images in the review process, some of the images look quite overexposed/bleached out in Figure 2., please check.

The Smc5/6 complex is important for HR – why does the nse4-4 allele show no increase in damage formation as measure by Rad52 foci formation? What temperature were these performed at?

What was the rationale for the use of two different sen1 mutants- sen1-1 vs sen1-3? Please provide details?

Please provide binding and dissociation constants to support the following statement: Together, our EMSA experiments indicate that the SMC5/6 complex can associate with R-loops with high affinity and specificity or , change the sentence to something like

"Together, our EMSA experiments indicate that the SMC5/6 complex associates with

R-loops with higher affinity relative to D-loops"

Given the results with purified proteins (Figure 6) – do the authors observe direct binding interactions (or not) between Smc5/6 and RNase H2? They have the reagents, so this experiment would be impactful as it would provide mechanistic insight about what Smc5/6 is doing to promote RNA-DNA hybrid metabolism. Do the author have any results describing Sumoylation (mms21 mutants) in the process?

*Reviewer #2 (Recommendations for the authors):*

Roy et al. explored the functional connections between RNase H and the Smc5/6 complex, using genetic and biochemical tools, and shedding light on a shared role in the metabolism of DNA-RNA hybrids. Authors provided evidence for a synthetic sick interaction between RNase H and Smc5/6 mutants, accompanied by an accumulation of hybrids, particularly evident at the rDNA and telomeres of double mutants. Additionally, the authors demonstrated the binding of the human Smc5/6 complex to DNA-RNA hybrids and its ability to stimulate RNase H2 activity in vitro. These discoveries point to a cooperation between RNase H2 and Smc5/6 in removal of RNA-DNA hybrids, offering valuable insights into the interplay between these two key factors in genome integrity.

One notable strength of the paper is the genetic interaction analysis between yeast Smc5/6 mutants and various pathways involved in hybrid metabolism, including RNase H genes, elements of the THO complex or the Sen1 helicase, which leads authors to conclude that Smc5/6 works mainly at highly transcribed regions, rather than at DNA replication-born DNA-RNA hybrids. Additionally, the authors successfully demonstrated the binding of the human Smc5/6 complex to DNA-RNA hybrids (with higher affinity than to DNA loops) and its capacity to enhance the activity of RNase H2 in vitro. The use of yeast cells and human complexes additionally underscores the evolutionary conservation of these mechanisms.

However, there are areas where the conclusions could benefit from further clarification of the presented data:

Specifically, the focus on combinations of mutations involving both RNase H1 and RNase H2 warrants exploration of genetic interactions when single RNase H1 or H2 genes are mutated in nse4-4 or smc6-9 mutants. A previous report (Lafuente-Barquero et al. 2017) showed that smc6-9 shows genetic interactions with RNase H2, but not with RNase H1. It would be pertinent to explore whether similar genetic interactions are observable when single RNase H1 or H2 genes are mutated in nse4-4 or smc6-9 mutants.

For additive effects between smc5/6 and hpr1/sen1-1/rat1-1 mutants, it would be important to test synergistic accumulation of hybrids in chromosome spreads. Also, detecting synthetic growth defects in pol2-MG or sen1-3 mutants is challenging since single mutants exhibit minimal phenotypes. This raises the possibility that they might function in the same pathway as Smc5/6 for R-loop removal. Addressing this question might be more effective by assessing hybrid accumulation in single and double mutants.

In terms of experimental methodology, the presence of hybrid foci outside the nuclei in chromosome spread experiments raises questions regarding their nature and inclusion in the analysis. It would be beneficial to give details about the nature of these foci and whether they are considered part of the nucleus. Standardizing the categorization of the number of foci per cell (e.g., no foci, 1 focus, 2 foci, etc.) across all figures would facilitate consistent analysis and comparison. This approach could also allow for a more careful evaluation of hybrids in single mutants. Additionally, it would be valuable to investigate whether changes in hybrid foci are primarily due to an increase in their number or also involve alterations in their size.

*Reviewer #3 (Recommendations for the authors):*

The study by Roy et al. introduces a novel function for the Smc5/6 complex, suggesting its involvement in the detection and degradation of R-loops formed during transcription. The authors demonstrate strong genetic interactions between the Smc5/6 complex and Rnase H2 mutants, leading to cell lethality. Mutations in both complexes impair cell growth, particularly at higher temperatures and in the presence of various DNA-damaging agents. The growth defects in double mutants are attributed to increased R-loop accumulation at chromosomes. The study concludes that the Smc5/6 complex exhibits a strong affinity for R-loop structures, recruiting the Rnase H2 complex for their degradation to maintain genome stability. This study sheds light on the role of chromatin architecture protein complexes, such as Smc5/6, in genome stability maintenance. While the genetic analysis supports the overall conclusion, further extension of genomic and in vitro data is recommended to strengthen the claims made by the authors. Particularly:

(i) The reliance solely on RNA:DNA hybrid foci using the S9.6 antibody for mapping R-loops presents a notable limitation in the study. Given that R-loops are the central focus of the paper, it would be beneficial for the authors to validate these findings through an alternative approach. Utilizing complementary methods such as DNA-RNA immunoprecipitation followed by sequencing (DRIP-seq) or bisulfite sequencing could strengthen the robustness of the R-loop mapping and enhance the credibility of the study's conclusions.

(ii) The article emphasizes the role of the SMC complex in highly transcribed regions and highlights only three loci, including one telomere and two loci from the rDNA region. However, to firmly establish the direct involvement of the SMC complex in recognizing and binding R-loops, it is imperative to conduct a comprehensive genome-wide analysis of R-loops. This analysis should be juxtaposed with the binding pattern of the SMC complex across the entire genome. Such an approach would provide a more holistic understanding of the interplay between the SMC complex and R-loops and elucidate their functional significance across different genomic regions.

(iii) The article briefly mentions DNA-damaging agents such as HU, MMS, and 4NQO, yet fails to explore their effects on R-loop accumulation and the ensuing consequences. It is crucial to investigate how these agents impact the formation and stability of R-loops, as well as their potential implications for genome integrity and cellular homeostasis. Integrating this information into the discussion would enrich the understanding of the dynamic interplay between DNA damage response pathways and R-loop biology, thereby broadening the significance of the study's findings.

(iv) While the authors utilize gel shift assays to demonstrate the binding of the SMC complex to R-loops, it is essential to address the specificity of these claims through comprehensive controls. For instance, considering the possibility of R-loop binding being a general property of ring-shaped complexes such as cohesin and condensin, it is imperative to incorporate appropriate controls to discern specific interactions. Implementing mutant versions of the SMC complex or competing with non-specific DNA substrates could help delineate the precise mechanisms underlying the observed binding events and bolster the validity of the conclusions drawn from the assay results.

It's essential to improve the readability and academic integrity of the article by minimizing the reliance on review articles as primary references. While review articles can provide valuable insights and overviews of a topic, they should not dominate the reference list, especially in the initial references. Having the first 12 references comprised solely of review articles raises concerns regarding the depth and originality of the research presented. Incorporating more primary research articles alongside review papers would enhance the credibility and rigor of the study by directly referencing the original sources of data and findings.

Some statements within the text lack proper references, such as the statement on page 17 regarding "Previous studies…". It's crucial to provide citations for such assertions to support the claims made and to allow readers to access the relevant literature for further context and verification. By including appropriate references, the article can uphold scholarly standards and ensure transparency in attributing information to its sources.

The redundancy observed in the first paragraph of the discussion, resembling content from the Introduction section, raises questions about the necessity of repeating introductory information. Repetitive content not only hampers readability but also fails to add substantial value to the discussion. It's important to streamline the Discussion section by focusing on novel insights, interpretations, and implications arising from the study's findings rather than reiterating introductory concepts. By avoiding unnecessary repetition, the article can maintain reader engagement and convey its message more effectively.

---

## [Author Response]

Essential revisions (for the authors):Please address the following essential revisions:i) Please provide genetic interaction studies of RNH1 and RNH2 (single mutants) with nse4-4 and smc6-9.

This suggestion has been implemented. Figure 1—figure supplement 1 Panel B in the revised manuscript shows genetic interactions between *rnh1Δ* and *rnh2Δ (i.e., rnh201Δ, rnh202Δ, rnh203Δ)* single mutants with *smc6-9* and *nse4-4*, respectively.

ii) For quantitation of S9.6 immunofluorescence (Figure 2A), it may serve the authors better to measure A.U. per nucleus spread rather than foci count (>10), since some of those signals appear to be clustered. Please also address how background foci are dealt with for this assay.

This suggestion has been implemented. Figure 2—figure supplement 1 shows the quantification of S9.6 immunofluorescence intensity (A.U.) across entire fields of view of spread nuclei. Reassuringly, the result obtained with S9.6 total intensity tracks nicely with our earlier results based on the number of nuclei containing >10 S9.6 foci per nuclei. Our initial microscopy results (Figure 2A) relied on a “foci per nuclei” method because it is one of the most frequently used approach to perform this type of quantification in the field (PMID: 32749214, PMID: 24743342, PMID: 35866610). We now show both modes of quantifications in the revised manuscript. As further reassurance, we also provide below a correlation analysis of the results obtained with both types of quantifications (Author response image 1). As can be seen from the figure, the two methods show strong positive correlation to each other.

**Author response image 1. sa2fig1:** Graph showing the correlation of results obtained when quantifying S9. 6 RNA-DNA hybrids signal by the “foci per nuclei” and “total fluorescence intensity” methods. (R-squared- 0.9314, p-value<0.0001, Pearson correlation test).

iii) Please provide S9.6 immunofluorescence experiments for interaction with THO complex and Sen1 (Figure 4).

The suggestion has been implemented. We have provided the results in Figure 4—figure supplement 1 of the revised manuscript. As expected, we observed significant increase in the S9.6 R-loop signal in *nse4-4 sen1-1* and *smc6-9 hpr1Δ* compared to the corresponding single mutants.

iv) Please provide quantitative (dissociation constant) measurements for EMSA experiments. Competitive titration experiments with D-loop will strengthen the claim.

The suggestion has been implemented. We provide the dissociation constant for the EMSA experiment in the revised Figure 5—figure supplement 1 of the manuscript.

v) Please include key primary references in the manuscript, where missing.

We have improved the referencing of the manuscript significantly. We now cite key primary research articles along with fewer review articles, as suggested by reviewer #3.

Reviewer #1 (Recommendations for the authors):The Authors characterize the combined impact of the Smc5/6 complex and RNase H in R-loop degradation. The work is solid and will provide an advance, but how Smc5/6 is functioning mechanistically remains obscure. The work identifies altered toxic hybrids when the smc5/6 complex is compromised however, it is difficult to distinguish between cause and effect. For example, is the in vivo localization of the Smc5/6 complex to regions of the genome altered by RNA-DNA hybrid levels?

We thank reviewer #1 for the positive assessment and constructive criticism on our study. We have addressed all the points raised by this reviewer. Please see below for further clarification.

Location of rnh201D and rnh203D are unclear from Figure 1A. I believe one is mislabelled.

The labelling has been corrected in the revised figure.

Has it been determined what rnh1D alone looks like with the smc6-9 and nse4-4?

Figure 1—figure supplement 1 B of the revised manuscript now shows the genetic interaction between *rnh1Δ* and *smc6-9* or *nse4-4,* respectively.

The authors state that the presence of S9.6 foci was highest in the nse4-4 rnh1Δ rnh201Δ and smc6-9 rnh1Δ rnh201Δ strains, consistent with the fact that rnh201Δ represents the deletion of the catalytic subunit of RNase H2 (Figure 2A). – How would this be interpreted in the context of rnh1Δ rnh201Δ double mutants alone, which show less hybrid formation than counterparts. Please consider rephrasing.

We have observed that the *rnh1Δ rnh201Δ* double mutant contains mainly 3-10 S9.6 foci per nuclei (and only very few numbers of nuclei with > 10 foci). While not being the highest level in our classification, this category of nuclei (i.e, containing 3-10 S9.6 foci) is abnormal and infrequently observed in wild-type cells. Importantly, Figure 2—figure supplement 1 A shows that *rnh1Δ rnh201Δ* have comparable numbers of nuclei with 3-10 foci compared to the rest of the double mutant. Hence, we believe our initial conclusion is supported by the data. However, we appreciate the suggestion made by the reviewer and have rephrased the statement in the manuscript to improve clarity (Page 7 Line 13).

Importantly, overexpression of RNase H1 largely suppressed the S9.6 signal on chromatin spreads (Figure 2B), indicating that the foci described above are reflective of RNA-DNA hybrid formation in double mutant strains. (p.6) – add more detail into how this was performed in the Results section – ectopic expression?

As per the suggestion, we have elaborated the text in the result section, for further details see Page 7 Line 18. Strains expressing RNase H1 were generated by integrating *YIplac204::P_GAL1_::RNH1* at the *TRP1* locus and integration was confirmed by PCR screening. The cells were induced by galactose to induce the overexpression of RNase H1.

The quantification of nse4-4 rnh1D rnh203D +RNH1 doesn't look representative of the microscopy. Also, please address why there more foci in nse4-4 rnh1D rnh203D+RNH1 compared to nse4-4 single mutants, shouldn't these be the same? Perhaps it is the images in the review process, some of the images look quite overexposed/bleached out in Figure 2., please check.

The suggestion has been implemented. We now show a better representation of the *nse4-4 rnh1Δ rnh203Δ +RNH1* in the updated Figure 2 panel B. As a technical side note, we would like to mention that overexpression of *RNH1* is not expected to completely compensate for the endogenous deletion of the two RNase H enzymes. This is because the cells still lack fully activated RNase H2 and it has been previously shown that the R-loop substrates for RNase H1 and RNase H2 does not fully overlap (PMID: 22244334, PMID: 31775053). Hence, *nse4-4 rnh1Δ rnh203Δ +RNH1* do not totally resemble *nse4-4.*

The Smc5/6 complex is important for HR – why does the nse4-4 allele show no increase in damage formation as measure by Rad52 foci formation? What temperature were these performed at?

The experiment was performed at 23 °C, a permissive temperature for *nse4-4*. We have added more details in the methods section of the revised manuscript **Page 25 Line 7**. As *nse4-4* is a temperature sensitive allele, we do not expect to see drastic damage formation or Rad52 foci in the absence of external (e.g., MMS, IR, NQO) DNA damage or heat shock treatment.

What was the rationale for the use of two different sen1 mutants- sen1-1 vs sen1-3? Please provide details?

Our aim was to test whether the Smc5/6 complex prevents R-loop formation across the entire genome or only at a specific subset of locations in the genome. The Sen1 helicase participates in R-loop prevention during DNA replication as well as during gene transcription. *sen1-1* is a temperature-sensitive variant that carries the amino-acid substitution G1747D in the helicase domain of Sen1, hence hampering the catalytic activity of the enzyme. On the other hand, *sen1-3* mutant bears mutations in the N-terminal domain of Sen1 helicase which interacts with the replisome. The *sen1-3* allele can only affect the interaction of replisome with Sen1 without affecting the role of Sen1 in transcription and its helicase activity. Hence, *sen1-3* allele is linked to RNA-DNA hybrids formation mostly during replication by faulty interaction of Sen1 helicase with the replisome (Appanah et al. 2020). We have provided more details explaining the rationale behind using the two separate mutants in the revised manuscript Page 9 Line 7 and Page 10 Line 5.

Please provide binding and dissociation constants to support the following statement: Together, our EMSA experiments indicate that the SMC5/6 complex can associate with R-loops with high affinity and specificity or , change the sentence to something like"Together, our EMSA experiments indicate that the SMC5/6 complex associates withR-loops with higher affinity relative to D-loops"

This suggestion has been implemented. We provide the dissociation constant for the EMSA experiment in the revised Figure 5—figure supplement 1 of the manuscript and edited the text on Page 11 from Line 12.

Given the results with purified proteins (Figure 6) – do the authors observe direct binding interactions (or not) between Smc5/6 and RNase H2? They have the reagents, so this experiment would be impactful as it would provide mechanistic insight about what Smc5/6 is doing to promote RNA-DNA hybrid metabolism. Do the author have any results describing Sumoylation (mms21 mutants) in the process?

As per the suggestion from the reviewer, we have now tested the binding interaction of purified SMC5/6 complex and RNase H2 by an in vitro pull-down assay and co-immunoprecipitation assay in yeast extracts. The results are shown in Figure 6—figure supplement 2. We see no detectable physical interaction connecting the Smc5/6 complex and RNase H2 enzyme.

We also report a synthetic genetic interaction between a sumoylation-defective *mms21* mutant (point mutation in the SP-RING domain) and *rnh1Δ rnh201Δ, rnh1Δ rnh202Δ, rnh1Δ rnh203Δ* and *sen1-1* mutants (revised Figure 1—figure supplement 1 C). As expected, the observed synthetic phenotype of the *mms21* RNase H double mutant is similar to that of double mutants involving *nse4-4* and *smc6-9* alleles.

Reviewer #2 (Recommendations for the authors):Roy et al. explored the functional connections between RNase H and the Smc5/6 complex, using genetic and biochemical tools, and shedding light on a shared role in the metabolism of DNA-RNA hybrids. Authors provided evidence for a synthetic sick interaction between RNase H and Smc5/6 mutants, accompanied by an accumulation of hybrids, particularly evident at the rDNA and telomeres of double mutants. Additionally, the authors demonstrated the binding of the human Smc5/6 complex to DNA-RNA hybrids and its ability to stimulate RNase H2 activity in vitro. These discoveries point to a cooperation between RNase H2 and Smc5/6 in removal of RNA-DNA hybrids, offering valuable insights into the interplay between these two key factors in genome integrity.One notable strength of the paper is the genetic interaction analysis between yeast Smc5/6 mutants and various pathways involved in hybrid metabolism, including RNase H genes, elements of the THO complex or the Sen1 helicase, which leads authors to conclude that Smc5/6 works mainly at highly transcribed regions, rather than at DNA replication-born DNA-RNA hybrids. Additionally, the authors successfully demonstrated the binding of the human Smc5/6 complex to DNA-RNA hybrids (with higher affinity than to DNA loops) and its capacity to enhance the activity of RNase H2 in vitro. The use of yeast cells and human complexes additionally underscores the evolutionary conservation of these mechanisms.

We appreciate the positive assessment of our manuscript by reviewer #2. We have addressed the points raised by this reviewer below

However, there are areas where the conclusions could benefit from further clarification of the presented data:Specifically, the focus on combinations of mutations involving both RNase H1 and RNase H2 warrants exploration of genetic interactions when single RNase H1 or H2 genes are mutated in nse4-4 or smc6-9 mutants. A previous report (Lafuente-Barquero et al. 2017) showed that smc6-9 shows genetic interactions with RNase H2, but not with RNase H1. It would be pertinent to explore whether similar genetic interactions are observable when single RNase H1 or H2 genes are mutated in nse4-4 or smc6-9 mutants.

The suggestion has been implemented. Figure 1—figure supplement 1 Panel B in the revised manuscript shows the genetic interaction between only *rnh1Δ (RNase H1) or rnh201Δ, rnh201Δ, and rnh203Δ* (RNase H2) with *smc6-9* and *nse4-4*, respectively.

For additive effects between smc5/6 and hpr1/sen1-1/rat1-1 mutants, it would be important to test synergistic accumulation of hybrids in chromosome spreads. Also, detecting synthetic growth defects in pol2-MG or sen1-3 mutants is challenging since single mutants exhibit minimal phenotypes. This raises the possibility that they might function in the same pathway as Smc5/6 for R-loop removal. Addressing this question might be more effective by assessing hybrid accumulation in single and double mutants.

The suggestion has been implemented. We have provided the results for *smc5/6* and *hpr1/sen1-1* mutants in Figure 4—figure supplement 1 of the revised manuscript. As expected, we observed a significant increase in the S9.6 R-loop signal in *nse4-4 sen1-1* and *smc6-9 hpr1Δ* compared to the respective single mutants.

In terms of experimental methodology, the presence of hybrid foci outside the nuclei in chromosome spread experiments raises questions regarding their nature and inclusion in the analysis. It would be beneficial to give details about the nature of these foci and whether they are considered part of the nucleus. Standardizing the categorization of the number of foci per cell (e.g., no foci, 1 focus, 2 foci, etc.) across all figures would facilitate consistent analysis and comparison. This approach could also allow for a more careful evaluation of hybrids in single mutants. Additionally, it would be valuable to investigate whether changes in hybrid foci are primarily due to an increase in their number or also involve alterations in their size.

We appreciate the reviewer's perceptive comment. We have added more details in the revised manuscript regarding the standards followed for quantification of the number of RNA-DNA hybrid foci per nucleus. We have also clarified this point in ‘Essential revisions Point II’ above. Importantly, while counting the number of nuclei with S9.6 specific foci, we only focussed on the blue DAPI (puff like structure) area as the nuclei of interest to quantify the S9.6 foci. Specifically, we aimed to quantify intact nuclei spread and we considered that DAPI (blue) “puffs” represents roughly intact nuclei with little or no broken DNA fragments. We also provided the number of nuclei with 3-10 foci in Figure 2—figure supplement 1 of the manuscript for categorization. Finally, we have shown quantification based on intensity of the S9.6 foci (A.U.) with respect to the whole field of view that consider all the foci outside of the nucleus. We would like to clarify that we are not counting cells by this procedure (see underlined text in paragraph above), as the images only represent nuclei spread on the slide. Overall, the different modes of analysis point towards a consistent conclusion (correlation provided in Reviewer Figure 1 above) that the strains with inactive RNase H and Smc5/6 complex activity have more S9.6 RNA-DNA hybrid signal compared to their corresponding single mutants.

We believe that investigating whether the changes in the hybrid foci are primarily due to an increase in their number or also involve alterations in their sizes would be intriguing (i.e., last sentence in reviewer paragraph above). We note, however, that the preparation of chromosome spreads requires physical “spreading” of nuclei on a slide, which is likely to impact the native morphology of foci. As such, it would be difficult to differentiate whether any changes in the morphology of foci represent a physiological feature of a specific mutant or an artifact of the spreading technique.

Reviewer #3 (Recommendations for the authors):The study by Roy et al. introduces a novel function for the Smc5/6 complex, suggesting its involvement in the detection and degradation of R-loops formed during transcription. The authors demonstrate strong genetic interactions between the Smc5/6 complex and Rnase H2 mutants, leading to cell lethality. Mutations in both complexes impair cell growth, particularly at higher temperatures and in the presence of various DNA-damaging agents. The growth defects in double mutants are attributed to increased R-loop accumulation at chromosomes. The study concludes that the Smc5/6 complex exhibits a strong affinity for R-loop structures, recruiting the Rnase H2 complex for their degradation to maintain genome stability. This study sheds light on the role of chromatin architecture protein complexes, such as Smc5/6, in genome stability maintenance. While the genetic analysis supports the overall conclusion, further extension of genomic and in vitro data is recommended to strengthen the claims made by the authors. Particularly:

We thank reviewer 3 for the insightful comments on our manuscript. The concerns raised below have been addressed in the revised manuscript.

(i) The reliance solely on RNA:DNA hybrid foci using the S9.6 antibody for mapping R-loops presents a notable limitation in the study. Given that R-loops are the central focus of the paper, it would be beneficial for the authors to validate these findings through an alternative approach. Utilizing complementary methods such as DNA-RNA immunoprecipitation followed by sequencing (DRIP-seq) or bisulfite sequencing could strengthen the robustness of the R-loop mapping and enhance the credibility of the study's conclusions.

We would like to emphasize that we did use an alternative approach to validate the accumulation of RNA-DNA hybrids in addition to the use of S9.6 antibody. Our alternative approach targets the single-stranded DNA region of R-loop structures by overexpression of activation-induced cytosine deaminase (AID) enzyme (PMID: 35704184). In this approach, detection of R-loops is directly linked to AID-induced hyperrecombination and Rad52 foci formation. We chose this approach because it is a validated method to assess R-loop formation without the use of the S9.6 antibody (PMID: 35704184). Importantly, this method confirmed the observations we obtained with the S9.6 antibody.

(ii) The article emphasizes the role of the SMC complex in highly transcribed regions and highlights only three loci, including one telomere and two loci from the rDNA region. However, to firmly establish the direct involvement of the SMC complex in recognizing and binding R-loops, it is imperative to conduct a comprehensive genome-wide analysis of R-loops. This analysis should be juxtaposed with the binding pattern of the SMC complex across the entire genome. Such an approach would provide a more holistic understanding of the interplay between the SMC complex and R-loops and elucidate their functional significance across different genomic regions.

While we recognize the holistic value of the genome-wide studies proposed by the reviewer, we respectfully disagree that they are imperative to firmly establish the direct involvement of the SMC complex in R-loop physiology. One key reason for this assessment is that it was previously shown that R-loop formation is highly enriched in genomic regions such as rDNA locus, telomeres, and highly transcribed/difficult to replicate chromosomal regions (PMID: 24743342, PMID: 29104020, PMID: 25357144, PMID: 27298336), and the Smc5/6 complex has been shown to accumulate in these same regions by chromatin immunoprecipitation (Pebernard et al., 2008, Lindroos et al., 2006.). Furthermore, the Smc5/6 complex plays a vital role in replication and segregation of repetitive rDNA region which are prone to form R-loops (Pebernard et al., 2008, Lindroos et al., 2006., Moradi Fard et al., 2021, Torres-Rosell et al., 2005). Similarly, the Smc5/6 complex plays an important role in telomeric length maintenance where Telomeric Repeat-containing RNA (TERRA) are commonly found (Moradi-Fard et al. 2016, Potts et al. 2007). Finally, recent studies showed that the Smc5/6 complex is enriched at transcription-induced positively supercoiled DNA and is linked to DNA topology management during transcription (Jeppsson et al., 2024). Hence, based on available data, one can conclude that there is a strong overlap of R-loop formation sites and Smc5/6 complex localization in the genome. Also, it is important to point out that no RNA-DNA hybrid metabolism regulator acts on all R-loop substrates formed in the genome. As a consequence, there are limits to the expectation that the localization pattern of the Smc5/6 complex (or any R-loop enzyme) should fully overlap with all R-loop formation sites on chromosomes.

(iii) The article briefly mentions DNA-damaging agents such as HU, MMS, and 4NQO, yet fails to explore their effects on R-loop accumulation and the ensuing consequences. It is crucial to investigate how these agents impact the formation and stability of R-loops, as well as their potential implications for genome integrity and cellular homeostasis. Integrating this information into the discussion would enrich the understanding of the dynamic interplay between DNA damage response pathways and R-loop biology, thereby broadening the significance of the study's findings.

We appreciate the point raised by the reviewer and believe the issue s/he raised has been addressed in the literature. Specifically, replication stress induced by HU or MMS has been shown to increase RNA-DNA hybrid abundance in the yeast genome, and this accumulation is exacerbated in RNase H mutant cells (PMID: 37855233; PMID: 31775053). Consistent with this, low dosage of HU leads RNase H mutant cells to mitotic checkpoint arrest causing massive cell lethality (PMID: 22244334). Additionally, treatment of budding yeast and human cells with 4-nitroquinoline-1-oxide induces polyubiquitylation of the largest RNA polymerase II subunit which affects overall transcription and transcription induced RNA-DNA hybrid formation (PMID: 16705154). We have modified the text of the revised manuscript (Page 3 Line 17 and Page 6 Line 12) to clarify this point.

(iv) While the authors utilize gel shift assays to demonstrate the binding of the SMC complex to R-loops, it is essential to address the specificity of these claims through comprehensive controls. For instance, considering the possibility of R-loop binding being a general property of ring-shaped complexes such as cohesin and condensin, it is imperative to incorporate appropriate controls to discern specific interactions. Implementing mutant versions of the SMC complex or competing with non-specific DNA substrates could help delineate the precise mechanisms underlying the observed binding events and bolster the validity of the conclusions drawn from the assay results.

We thank the reviewer for these excellent suggestions, and we have performed the suggested experiments to strengthen our study. To test the R-loop binding ability of another ring-shaped complex, we purified yeast condensin in monomeric and multimeric forms (PMID: 29079757) and prepared R-loop substrates to conduct binding experiments by electrophoretic mobility shift assays (EMSAs). We observed that monomeric SMC5/6 complex binds R-loop substrates with very high efficiency but similar concentrations of monomeric condensin fails to bind R-loop substrates. In contrast, we observed effective R-loop binding with the oligomeric condensin complex. These results demonstrate that binding to R-loop structures is not a universal feature of all ring-shaped complexes (i.e., as seen with the monomeric condensin result). These new results have been included in Figure 5—figure supplement 2 of the revised manuscript.

We have also performed the DNA binding/competition experiment requested by the reviewer above. Specifically, to evaluate the specificity of the SMC5/6 complex binding to R-loop (compared to D-loop), increasing concentrations of human SMC5/6 complex were incubated with 100x molar excess of unlabeled poly-deoxy-inosinic-deoxy-cytidylic acid (poly[d(I-C)]) along with 40 nM of [^32^P]-labelled R-loop or D-loop followed by EMSA analysis. The results of this experiment fully support the conclusions presented in our original manuscript and are shown in the updated Figure 5 of the revised manuscript.

It's essential to improve the readability and academic integrity of the article by minimizing the reliance on review articles as primary references. While review articles can provide valuable insights and overviews of a topic, they should not dominate the reference list, especially in the initial references. Having the first 12 references comprised solely of review articles raises concerns regarding the depth and originality of the research presented. Incorporating more primary research articles alongside review papers would enhance the credibility and rigor of the study by directly referencing the original sources of data and findings.

We thank the reviewer for this helpful comment. We now cite several additional primary research articles in the revised manuscript to enhance the credibility and rigor of the study.

Some statements within the text lack proper references, such as the statement on page 17 regarding "Previous studies…". It's crucial to provide citations for such assertions to support the claims made and to allow readers to access the relevant literature for further context and verification. By including appropriate references, the article can uphold scholarly standards and ensure transparency in attributing information to its sources.

We appreciate the suggestion, and we have diligently improved the referencing in the revised manuscript.

The redundancy observed in the first paragraph of the discussion, resembling content from the Introduction section, raises questions about the necessity of repeating introductory information. Repetitive content not only hampers readability but also fails to add substantial value to the discussion. It's important to streamline the Discussion section by focusing on novel insights, interpretations, and implications arising from the study's findings rather than reiterating introductory concepts. By avoiding unnecessary repetition, the article can maintain reader engagement and convey its message more effectively.

We appreciate the reviewer’s comment. We have now edited the beginning section of the discussion on Page 14 line 1 of the updated manuscript to avoid redundancy and improve the readability of the discussion.